# TEST-TIME TRAINING DONE RIGHT

**Tianyuan Zhang**[1]   **Sai Bi**[2]   **Yicong Hong**[2]   **Kai Zhang**[2]   **Fujun Luan**[2]
**Songlin Yang**[1]   **Kalyan Sunkavalli**[2]   **William T. Freeman**[1]   **Hao Tan**[2]

[1]Massachusetts Institute of Technology     [2]Adobe Research

## ABSTRACT

Test-Time Training (TTT) models context dependencies by adapting part of the model's weights (often referred to as fast weights) at inference time. This adapted fast weight, similar to recurrent states in RNNs, stores temporary memories of past tokens in the current sequence. Existing TTT methods have struggled to demonstrate effectiveness in handling long-sequence data, due to their computational inefficiency on modern GPUs. The TTT layers in many of these approaches operate with extremely low FLOPs utilization on modern GPUs(often below 5%) because they deliberately apply small online mini-batch sizes (e.g., updating fast weights every 16 or 64 tokens). Moreover, a small mini-batch implies fine-grained block-wise causal dependencies in the data, making them unsuitable for data beyond 1D ordered sequences, like sets or N-dimensional grids such as images or videos. In contrast, we pursue the opposite direction by proposing an extremely large chunk update, ranging from 2K to 1M tokens across tasks of varying modalities, which we refer to as Large Chunk Test-Time Training (LaCT). This approach improves hardware utilization by orders of magnitude, and more importantly, facilitates scaling of nonlinear state size (up to 40% of model parameter size), hence substantially improving state capacity, all without requiring cumbersome and error-prone custom kernel implementations. It also allows easy integration of sophisticated optimizers like Muon for online memory updates. We validate our approach across diverse data modalities and tasks, including novel view synthesis from image sets, language models, and auto-regressive video diffusion models. Our approach can scale up to 14-billion-parameter auto-regressive video diffusion models handling sequences of up to 56K tokens. In our longest sequence experiment, we perform novel view synthesis with more than one million context length. Our results highlight the computational and performance benefits of large-chunk test-time training, paving the way for more efficient and scalable long-context sequence modeling. We hope that this work will inspire and accelerate new research in the field of long-context modeling and test-time training. See visual results on project website `https://tianyuanzhang.com/projects/ttt-done-right/`.

## 1 INTRODUCTION

The demand for handling long contexts is rapidly growing. While softmax attention (Vaswani et al., 2017) has become the de facto solution for modeling various types of data, its computational cost grows quadratically with sequence length, motivating extensive research into more efficient long-context modeling.

Recently, Test-Time Training (TTT) (Sun et al., 2024) has emerged as a promising approach for efficient sub-quadratic sequence modeling. TTT extends the concept of recurrent states in RNNs to a small, online-adapted sub-network. The parameters of this sub-network also referred to as fast weight (Schlag et al., 2021), as they are rapidly adapted online via self-supervised objectives to memorize in-context information. Numerous recent studies (Wang et al., 2025b; Behrouz et al., 2024; 2025b; Karami & Mirrokni, 2025) have explored various online objectives, optimizers, and architectures for fast weight networks.

Despite these efforts, existing TTT methods struggle to scale effectively to long contexts, primarily due to extremely low hardware utilization in their TTT layers (often below 5% peak FLOPS on

modern GPUs). This inefficiency is because of the usage of small mini-batch sizes, i.e. updating fast weights every token or every 16 to 64 tokens, which is conventionally assumed to be more effective for in-context learning. Such small mini-batch results in poor parallelism and low compute intensity, and presents significant challenges for hardware-efficient implementation, especially when using large, nonlinear fast weights, making it difficult to achieve non-trivial (above 10%) FLOPs utilization.

In this paper, we adopt the opposite strategy and introduce Large Chunk Test-Time Training (LaCT). LaCT leverages extremely large chunk (from 2048 to 1M tokens) as the basic unit to update the fast weight. Since the tokens within each large chunk are treated as an unordered set, we further integrate window attention into LaCT to capture local dependencies within the chunk. LaCT significantly enhances parallelism, leading to substantially improved GPU utilization (up to 70% on NVIDIA A100s) with just a few dozen lines of pure PyTorch code (see Appendix C.1). This efficiency enables the scaling of non-linear fast weights to enhance the memory capacity. And simple implementation allows easy integration of more effective test-time optimizers, such as Muon (Jordan et al., 2024).

Furthermore, LaCT's large-chunk design is also natural to model diverse N-dimensional data as we can align chunk-size with the internal structure of the data (e.g., grouping tokens within an image or consecutive video frames as a chunk).

We extensively validate LaCT on three tasks spanning different modalities and data structures:

- *Novel View Synthesis.* Our model is capable of processing up to $128$ input images at a resolution of $960 \times 536$ leading to a maximum of 1M tokens, and outperforms 3D Gaussian Splatting (Kerbl et al., 2023) in terms of rendering quality under such input scale.

- *Language Modeling.* Our model achieves competitive performance compared to SoTA methods such as DeltaNet (Yang et al., 2024b), even though a chunk structure is not explicitly present in language data.

- *Autoregressive Video Diffusion.* We adapt a 14-billion-parameter bidirectional video diffusion transformer into an autoregressive model by incorporating LaCT with sliding window attention. This adapted model generates consistent videos up to 56,000 visual tokens.

To summarize, our approach establishes an efficient, scalable, and highly performant framework for long sequence modeling across diverse modalities. By removing the dependency on low-level, hardware-specific implementations, LaCT enables broader exploration of the architectural design space. We believe this can democratize research in efficient long-context modeling and inspire the development of more novel and effective designs.

## 2 RELATED WORK

**Test-time training.** Test-Time Training (TTT) (Sun et al., 2024) is an emerging concept in sequence modeling that extends the concept of recurrent states in RNNs to online-adapted neural network components. In TTT models, a subset of weights, termed "fast weights," are updated to learn in-context. Existing methods typically employ a self-supervised loss that encourages these fast weights to memorize key-value associations from in-context tokens, using variants of gradient descent for online adaptation. TTT (Sun et al., 2024; Wang et al., 2025b) has opened a vast design space for new recurrent model architectures. For instance, many recent works have developed novel test-time optimizers (Behrouz et al., 2024; Karami & Mirrokni, 2025; Behrouz et al., 2025a) and online training objectives (Behrouz et al., 2025b). However, current nonlinear TTT approaches often suffer from low hardware utilization and limited state sizes, and consequently have not yet demonstrated their full potential. Our work primarily addresses these challenges by advocating for a new paradigm of using extremely large online minibatch (chunk) sizes for updating the fast weights. This paradigm can achieve orders-of-magnitude higher hardware utilization without relying on error-prone custom kernel implementations. Furthermore, it enables efficient scaling of nonlinear state sizes and offers the flexibility to use diverse fast weight neural networks and optimizers, thereby accelerating research progress in this area.

**Combining chunk attention with recurrence.** Several recent models combine local chunk attention with linear recurrence, such as Gated Attention Unit (GAU) (Hua et al., 2022b), MEGA (Ma et al., 2023), MEGALODON (Ma et al., 2024), and InfiniAttention (Munkhdalai et al., 2024).

Among these, InfiniAttention is conceptually closest to our work, as it incorporates recurrence at the chunk level using the delta rule—interpreted as an online linear regression objective from the perspective of Test-Time Training (TTT). However, this update rule is limited in expressivity. In contrast, we employ a significantly more expressive update mechanism derived from a more general TTT framework, and demonstrate the substantial gains this brings.

Block-Recurrent Transformer (Hutchins et al., 2022) also explores large chunk memory updates, where memory tokens act as recurrent states that can self-attend and cross-attend with input tokens during each chunk update via attention mechanisms. The Perceiver-style register-token attention baseline used in our novel view synthesis experiments (Sec. 6.1, Table 2) is conceptually similar to the Block-Recurrent Transformer in its use of register tokens for context compression. As shown in Figure 4, our method significantly outperforms this approach in both speed and quality, with a comparable state size.

Further related work on novel view synthesis and video generation is discussed in Appendix B.

## 3 PRELIMINARY

### 3.1 TEST-TIME TRAINING

Consider a one-dimensional sequence of $N$ tokens $\mathbf{x} = [x_1, x_2, \ldots, x_N]$, where each token $x_i \in \mathbb{R}^d$. Following attention formulation, each input tokens $x_i$ is projected into query ($q_i$), key ($k_i$), and value ($v_i$) vectors. For clarity, we assume all these vectors $q_i, k_i, v_i \in \mathbb{R}^d$.

Test-Time Training (TTT) (Sun et al., 2024) introduces a neural network with rapidly adaptable weights—called *fast weights* (Hinton & Plaut, 1987; Schmidhuber, 1992; Schlag et al., 2021)—that are updated during both training and inference to dynamically store context information. This contrasts with the *slow weights* (i.e., model parameters) that are frozen during inference. Formally, TTT defines fast weights in the form of a neural network: $f_W(\cdot) : \mathbb{R}^d \to \mathbb{R}^d$ parameterized by the fast weights $W$, and it involves two primary operations:

$$\textbf{Update operation:} \quad W \leftarrow W - \eta \nabla_W \mathcal{L}\big(f_W(k), v\big) \tag{1}$$

where $\mathcal{L}(\cdot, \cdot)$ is a loss function between the transformed key $f_W(k)$ and the value $v$, commonly Mean Squared Error, designed to encourage the network to associate keys with corresponding values. $\eta$ is the learning rate. Intuitively, this learning objective is to encode the KV cache into a neural memory with fixed state size as *accurate* as possible (Wang et al., 2025b).

$$\textbf{Apply operation:} \quad o = f_W(q), \tag{2}$$

where the updated fast weights $W$ are used to compute the output vector $o$ given the query $q$. The per-token TTT layer iteratively perform the update and apply operations on each token $x_i$ in sequence.

### 3.2 CHALLENGES IN EFFICIENT IMPLEMENTATION

Frequent online update of fast weights is inefficient due to memory bandwidth limitations. Consequently, previous works (Sun et al., 2023; Gu & Dao, 2023; Yang et al., 2024a; Qin et al., 2024; Yang et al., 2024c) often employ customized kernels that keep fast weights in SRAM across updates to reduce memory load. However, this strategy typically requires fast weights to evolve mostly independently within SMs to reduce communications, which is not valid for large nonlinear states (e.g., the nonlinear SwiGLU fast weight in Sec. 4.1 and the Muon update in Sec. 4.2). Moreover, developing such kernel code is cumbersome, with far longer development cycles than native PyTorch code, hindering rapid research exploration.

On the other hand, a PyTorch-based implementation, while simpler, is typically bounded by memory speed. As an illustration, consider a PyTorch implementation of simple MLP fast weight, the core of which is a matrix multiplication between fast weight (e.g., $h \times h$ matrix) and the mini-batch input ($b \times h$ where b is the chunk size). The ideal compute-to-memory ratio is:

$$r = \frac{2h^2 b}{2h^2 + 4hb} = \frac{h/2}{1 + \frac{h}{2b}} = \frac{b}{1 + \frac{2b}{h}} \leq \min(h/2, b). \tag{3}$$

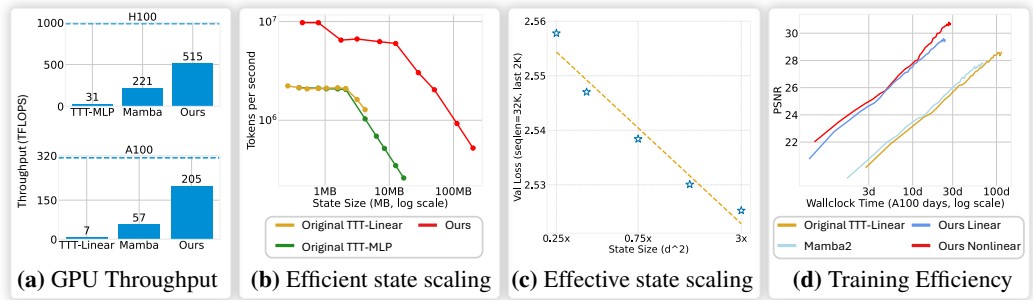

(a) GPU Throughput    (b) Efficient state scaling    (c) Effective state scaling    (d) Training Efficiency

Figure 1: Using larger chunk sizes significantly improves GPU utilization compared to the original test-time training (TTT) method that even uses customized kernels **(a)**. This enhanced utilization enables efficient and effective scaling to larger state sizes **(b)**, **(c)**, leading to better overall performance in less wall-clock time **(d)**. The dotted line in **(a)** is the theoretical peak BF16 throughput of the GPU. Panel **(c)** measure average validation loss of the last 2K tokens in sequences processed by a LaCT language model across varying state sizes, demonstrating benefits of larger state size. Panel **(d)** compares performance versus training time across different baselines on the novel view synthesis benchmark. Further experimental details can be found in Appendix E.4.

Here, $2h^2b$ is the FLOPs to for matrix multiplication, the denominator $2h^2 + 4hb$ is the memory workload for two input matrices and the output in BF16 (2 bytes). Small fast weight size (e.g., $h = 64$) or small chunk size (e.g., $b = 16$) will bound the ratio $r$ far below the theoretical peak (e.g., 290 FLOPs per byte on H100), making the operation memory-bound and limiting compute usage.

In light of this, we advocate for using large chunk sizes (from 2048 to 1M). This allows us to achieve higher throughput (Fig. 1a) leading to better performance in less training wall-clock time(Fig. 1d). Our design also allows the state size to be scaled up efficiently(Fig. 1b), leading to significant results improvement with such scaling (Fig. 1c, Fig. 7a). Our architecture achieves a state-to-parameter size ratio $\geq 40\%$, which is an order of magnitude larger than previous methods' ratio of $0.1\%$ to $5\%$. Detailed pseudocode is provided in Alg. 1.

**Parallelism over the sequence length dimension**, in addition to the batch and head dimensions, is crucial to achieve high occupancy when handling long sequences (where the batch size is often small). Linear Attention variants like Mamba (Gu & Dao, 2023), Gated Linear Attention (Yang et al., 2024a) and DeltaNet (Yang et al., 2024c) enable such parallelism by utilizing the associative property of linear recurrence. Attention (Vaswani et al., 2017) can be parallelized along the sequence length dimension due to independent computation between different queries. Supporting such parallelism is a key improvement of FlashAttention-2 (Dao, 2023) over FlashAttention-1 (Dao et al., 2022). For test-time training with non-linear updates, sequence dimension parallelism can only be implemented within online chunks, further motivating the use of extremely large chunk sizes. When implementing large-chunk TTT with PyTorch, this sequence dimension parallelism within a device across multiple thread blocks is automatically handled by PyTorch and low-level compilers. An example of such sequence parallelism across multiple devices is provided in Section 4.4, with pseudocode in Alg. 3.

## 4 LaCT MODEL ARCHITECTURE

As in Fig. 2, LaCT block consists of three types of layers: a window attention layer, a large-chunk TTT layer, and a feed-forward layer. Each layer is equipped with residual connections (He et al., 2015) following the practice in Transformer (Vaswani et al., 2017). The window attention layer performs local self-attention to capture the local dependency. In the TTT layer, we split the sequence into large chunks. The history context is gradually compressed into the fast weights through an 'update' operation (regarding vector key $K$ and value $V$), and latest weight is 'applied' to the current query vector (Q) to compute the output. The feed-forward layer performs channel mixing as in Transformer. We omit several linear and normalization layers in Fig. 2 for clarity and details are in Appendix C.1. Our framework offer great flexibility in handling diverse data types. In this section, we present general designs in our approach and later describe data-specific variations in Sec. 5.

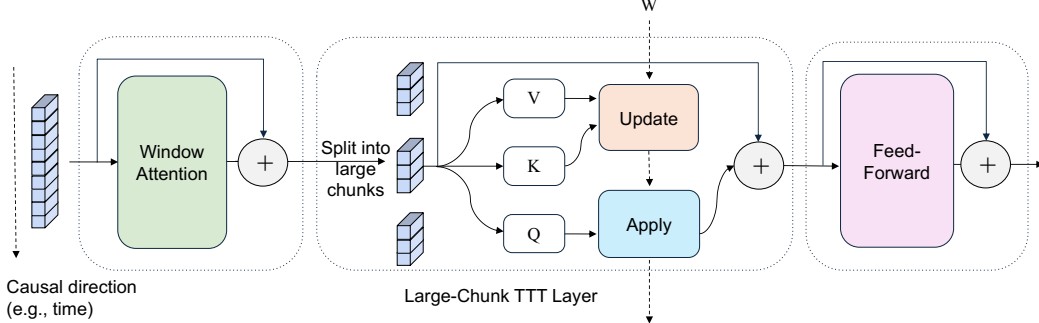

Figure 2: The basic diagram for a LaCT block. The large-chunk TTT layer updates the fast weight $W$ to store historical context information, while the window attention handles the locality and internal structures within the chunk. The solid line denotes the information flow over model depth and the dashed line denotes the information flow over time (i.e., the fast weight $W$ passing through chunks). Various instantiations in Sec. 5 use different chunk sizes and window attention types according to the specific data structure. Additionally, window attention and large-chunk TTT layers can be combined within the same layer by sharing the QKV and summing their outputs; this in-layer mixing is used in our language modeling and video generation experiments (see Alg. 2 for such pseudocode).

## 4.1 Large-Chunk TTT Layer

Different from the per-token update in Eqn. 1, the chunk-wise update computes the gradient of the summed loss over all keys $\{k_i\}$ and values $\{v_i\}$ within the chunk. As the chunk size is large, weight updates are performed infrequently. This enables more sophisticated weight-update rule designs (discussed in Sec. 4.2) and amortizes the update cost. The 'update' operation for the fast weight is:

$$g = \nabla_W \sum_{i=1}^{b} \eta_i \mathcal{L}\big(f_W(k_i), v_i\big) \tag{4}$$

$$W \leftarrow \text{weight-update}(W, g), \tag{5}$$

where $b$ is the chunk size, $g$ is the gradient of the fast-weight loss function, and $\eta_i$ is the learning rate of each token (usually predicted from input tokens). The 'apply' operation $o_i = f_W(q_i)$ is the same as Eqn. 2 and all query vectors $\{q_i\}$ in the chunk share the same updated fast weight $W$.

Motivated by recent LLMs (Touvron et al., 2023), we adopt SwiGLU-MLP (Shazeer, 2020) without bias terms as the fast-weight network. Our fast weights consists of three weight matrix $W = \{W_1, W_2, W_3\}$, and the network is:

$$f_W(x) = W_2 \left[\text{SiLU}(W_1 x) \circ (W_3 x)\right] \tag{6}$$

where $\circ$ is an elementwise multiplication. We apply a simple dot product loss as our loss function:

$$\mathcal{L}\big(f_W(k_i), v_i\big) = -f_W(k_i)^\top v_i \tag{7}$$

**Execution orders for 'apply' and 'update'.** Note that the 'update' operation and 'apply' operation of TTT are decoupled, and we can set the chunk size adaptively and apply these operation in different orders; this allows us to model diverse kinds of data dependencies, similar to different attention masks in self-attention. Figure 3 illustrates this concept. In Figure 3a, when the chunk size equals the full sequence length, performing the apply followed by the update operation is conceptually similar to full attention. Using update and apply alternately leads to a block-wise causal mask (Fig. 3b), where the block size corresponds to the chunk size. Switching the order between the two operations results in the a shift in the mask (Fig. 3c). This shifted mask does not leak future information within the chunk and is important when building the full causal mask in Language Modeling (Sec. 5.2). Moreover, only updating on a subset of chunks and applying to all (Figure 3d) is analogous to strided block-wise causal mask.

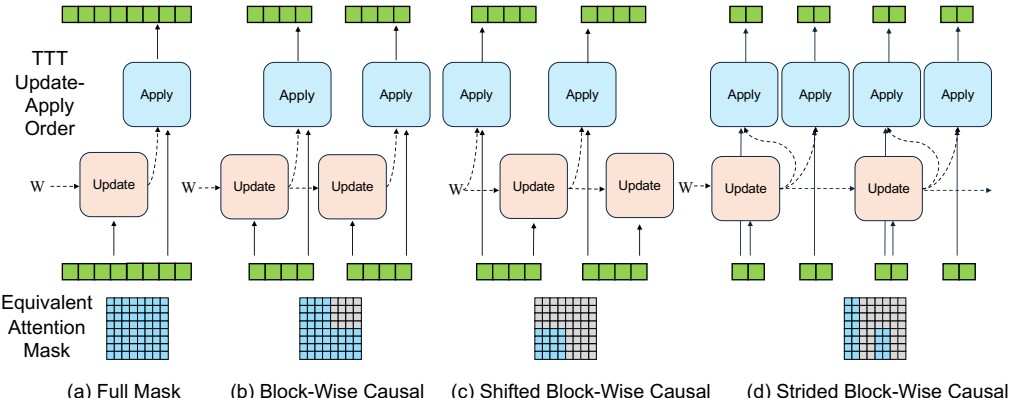

Figure 3: Different 'Update' and 'Apply' orders and their equivalent attention mask. A blue mask in i-th row and j-th column means the i-th token's output depends on the j-th token.

## 4.2 NON-LINEAR UPDATE OF FAST-WEIGHT

Fast-weight updates in TTT repeatedly accumulate gradients, and thus suffer from magnitude explosion or decayed memory. Large-chunk TTT allows non-linear updates to improve stability and effectiveness while preserving efficiency. For the 'weight-update' operation in Eqn. 5, our vanilla implementation involves gradient descent followed by weight normalization:

$$\text{weight-update}(W, g) = \text{L2-Normalize}(W - g). \tag{8}$$

We have also explored a more robust nonlinear Muon (Jordan et al., 2024) update rule [1] with weight normalization:

$$\text{weight-update}(W, g) = \text{L2-Normalize}(W - \text{Muon}(g)) \tag{9}$$

**Fast-weight normalization.** We apply L2 weight normalization to the updated fast weights along the input dimension. We do not use explicit weight-decay term as in previous methods (Behrouz et al., 2024; Dao & Gu, 2024; Yang et al., 2024a; Sun et al., 2023). When the network is conceptually rotated 90 degrees, treating the sequence dimension as the depth of a virtual model, the test-time training updates act as residuals over time (He et al., 2015). In this view, our fast-weight normalization is analogous to the *post-layer norm* in Transformer architectures, which constrains activation scales within the residual path.

**Muon-update rule.** Essentially, Muon normalizes the spectral norm of matrix gradient using Newton-Schulz iterations. In short, let $g = USV^T$ be the Singular Value Decomposition(SVD) of the gradient $g$, then Muon operator approximately converts the gradient as:

$$\text{Muon}(g) \simeq UV^T \tag{10}$$

Muon also improves the numerical stability in our setup. For example, the learning rate ($\eta_i$ in Eqn. 4) now only reflects the relative importance of tokens within a chunk as Muon normalizes the absolute scale. See (Jordan et al., 2024) and Appendix C for analysis of its computational cost.

## 4.3 WINDOW ATTENTION

The large-chunk TTT layer treats data as sequences of sets because its fast weight updates inherently disregard token order and spatial locality within each chunk. However, many data modalities—such as videos (sequences of grids), image collections (sets of grids), or text (1D sequences)—do not fully align with this set-based perspective . For these modalities, intra-chunk structure and locality are vital for capturing the overall data structure. We therefore integrate local window attention (either causal or bidirectional) alongside TTT layers to handle data structure within a chunk. Moreover, window attention efficiently handles localities in the data, enabling the TTT layer to focus its fixed-size fast weight capacity on modeling non-local dependencies. This hybrid strategy is also

---

[1]Muon requires weights in matrix form, and our current fast-weight function SwiGLU-MLP has three matrices as the weights (i.e., $W_1, W_2, W_3$ in Eqn. 6).

employed in other notable works like BASED (Arora et al., 2024), GAU (Hua et al., 2022a) and InifinitAttention (Munkhdalai et al., 2024). In summary, LaCT is a hybrid architecture with the quadratic-compute attention for local structure and linear-compute TTT for non-local context.

## 4.4 Context Parallelism

Context Parallelism (CP) partitions the sequence along the context length dimension and distributes the shards across multiple devices for parallel computing. The feed-forward layer and window attention are local operators thus natively support CP. For TTT layer, small chunks hardly support CP thus tensor parallelism (i.e., parallel over the heads) is preferred. Our large-chunk TTT layer allows CP by sharding the tokens within a chunk. This can be implemented through distributed all-reduce-sum and is logically the same as Distributed Data Parallelism (DDP), except that the parameters are the fast weights and input data are the tokens in the chunk. We adopt such parallelism in training the novel view synthesis task (Sec. 5.1) and observe minimal throughput overheads (1% to 3%). See Appendix for pseudocode (Alg. 3) and other parallelism(Alg. 4).

## 5 LaCT for N-Dimensional Data

In this section, we introduce the three tasks we address using LaCT—novel view synthesis, language modeling, and autoregressive video generation. These tasks have different inherent data structures and we address them with corresponding design choices. The full model architecture details for these data types are provided in Appendix D.

### 5.1 Novel View Synthesis - Image Set

Novel view synthesis (NVS)(McMillan & Bishop, 1995; Levo & Hanrahan, 1996) aims to render images of a static scene from previously unseen viewpoints. Formally, given a set of $N$ input posed images $\{(I_i, P_i)\}_{i=1}^N$ of a static scene, where $I_i \in \mathbb{R}^{H \times W \times 3}$ is an RGB image and $P_i$ is its corresponding camera pose, the model needs to synthesize new images from novel camera poses that typically do not overlap with the input views.

We find that NVS is an effective test bench for evaluating a model's online *memory* and *compression* capabilities. Firstly, NVS is challenging as it requires spatial compression, dense retrieval, and basic physical reasoning. Secondly, NVS can be formulated as a non-generative task, significantly reducing training computation and the need for extensive model parameters to store world knowledge, thereby enabling rapid experimentation. Thirdly, the substantial redundant information in dense input views incentivizes the model to learn effective compressions. Given these observations, we use NVS for our initial research iterations. We find that some of the insights gained are transferrable to other tasks.

Our NVS model follows the basic LaCT diagram in Sec. 4. Both the posed input images and poses of the target novel views are tokenized by patchify and linear layers, following LVSM (Jin et al., 2024a). The window attention exactly covers the tokens from a single image. The LaCT layer adapts a single-round of strided block-wise causal mask (Fig. 3d), which updates the fast weight using all input view tokens, and applies to both the input and novel view tokens. The *update* step resembles a prefill stage, while the *apply* operation resembles parallel decoding. During rendering of novel views, each test-time training layer functions as a static weight layer, making the entire model a static vision transformer (Dosovitskiy et al., 2020). We illustrate this design in Figure 9.

### 5.2 Language Modeling - Text Sequence

Autoregressive language models predict the probability distribution of the next token given preceding tokens, $p_\theta(x_n | x_1, \ldots, x_{n-1})$. Text sequences lack inherent chunk structures, so for LaCT, we define chunk size as a hyperparameter (e.g., 2048 or 4096 tokens). We utilize the shifted block-wise causal mask as in Fig. 3(c) for the TTT apply-update sequence to avoid seeing future tokens in a chunk. Since LaCT lacks per-token causality within each chunk, we employ sliding window attention—with window size equal to the chunk size—to efficiently model per-token causal dependencies. The sliding window is integrated into the same TTT layer with shared QKV similar to GAU (Hua et al., 2022a). We illustrate the detailed architecture in Fig. 10 and pseudocode 2.

Table 1: Summary of our experiments on three different data structures. 'd' denotes model dimension. The state size denotes the size of the fast weight per model block.

| Task name | Data Structure | Chunk Size | State Size | Model Size | Max Length | Context Parallelism |
|---|---|---|---|---|---|---|
| Novel View Synthesis | Image set | Full sequence | $6d^2$ | 0.3B | 1M | Within-chunk parallel |
| AR Video Diffusion | Image sequence | Three frames | $3d^2, 0.75d^2$ | 1.3B, 14B | 56160 | Head-dim parallel |
| Language Models | 1D Sequence | 2K, 4K tokens | $0.75d^2$ | 0.7B, 3B | 32768 | N/A |

## 5.3 AUTOREGRESSIVE VIDEO DIFFUSION - IMAGE SEQUENCES

Chunkwise autoregressive video diffusion iteratively denoises a number of subsequent video frames, conditioned on the previously generated clean frames, where each chunk can contain thousands of visual tokens. We use teacher-forcing training by interleaving noisy and clean frame chunks. Specifically, a video of N frame chunks is structured as:

$$S = [X_1^{\text{noise}}, X_1, X_2^{\text{noise}}, X_2, \dots, X_N^{\text{noise}}] \tag{11}$$

where each noisy chunk $X_i^{\text{noise}}$ is produced by adding unit Gaussian noise $\epsilon$ to the $i$-th clean video chunk as $X_i^{\text{noise}} = X_i(1 - t_i) + \epsilon t_i$ and $t_i \in [0, 1]$ denotes the strength of chunk-independent noise.

To handle such a data structure, we employ the strided block-wise causal mask in Fig. 3d for LaCT. Specifically, it *applies* fast weights to each chunk sequentially while only *updating* fast weights on clean chunks. This simple strategy ensures that each denoising operation only accesses previously cleaned frames. The windowed attention uses a non-overlapping window with 2 consecutive chunks (i.e., $[X_i, X_{i+1}^{\text{noise}}]$) to build temporal and spatial locality. Within each window, the attention from $X_i$ to $X_{i+1}^{\text{noise}}$ is excluded. We adopt all attention and TTT masking patterns similar to Fig. 3c. The details of this hybrid architecture and efficient trainings are in the Appendix D.3.

## 6 EXPERIMENTS

In this section, we present our experiment results on novel view synthesis (Sec. 6.1), language modeling (Sec.6.2), and autoregressive video generation (Sec. 6.3), and an in-depth analysis (Sec. A) of different design choices. Tab. 1 summarizes key factors in each experiment. When comparing with linear-cost baselines, we augmented them with the same window attention for fair comparisons. The full experimental details for all tasks are provided in Appendix E.

### 6.1 NOVEL VIEW SYNTHESIS

**Datasets & metric.** We evaluate our method on both object-level (Deitke et al., 2023) and scene-level (Ling et al., 2024) datasets. For object-level experiments, models are trained on Objaverse and tested on the Google Scanned Objects (GSO) dataset (Downs et al., 2022), at $256 \times 256$ and $512 \times 512$ resolutions. Each evaluation uses 4–48 input views and 8 novel views per object. For scene-level experiments, we adopt 140 test scenes from DL3DV, evaluated at $960 \times 536$ resolution. Performance is reported in Peak Signal-to-Noise Ratio (PSNR), with additional metrics provided in Appendix E.1.

**Model & Training details.** Our default model has 312M parameters, including 84M fast weights ($6d^2$ per block). We train on 1.25T tokens for object-level datasets and 1.8T tokens for scene-level datasets, using a maximum sequence length of one million tokens. High-resolution models employ inner-chunk context parallelism (Sec. 4.4). More details are provided in Appendix E.1.

**Baselines.** For object-level evaluation, we compare against two representative attention designs: one without in-context compression and one with compression: (1) the non-compressed baseline is a full-attention model, where TTT layers are replaced by block-wise causal attention enabling bidirectional input interactions and cross-attention from novel views, and (2) the attention-with-compression baseline is a perceiver-style register-attention model (Jaegle et al., 2021), which compress inputs to 4096 registers and decodes novel views via cross-attention. For scene-level evaluation, we benchmark against LongLRM (Ziwen et al., 2024), which integrates Mamba (Gu & Dao, 2023) with full attention for 3D Gaussian splat prediction, as well as pure optimization-based 3D Gaussian splatting methods. Table 2 summarizes computational complexity across all models. Figure 4 present experimental results and analysis. See more results and details in Appendix. E.1.

Table 2: Complexities of methods on novel view synthesis w/ $n$ input. Prefill and rendering speed are measured on A100 with 48 512×512 input images (196K input tokens, 4K decoding tokens).

|  | State Size | Prefill Compute | Decoding Compute | # Params | Prefill speed | Rendering FPS |
|---|---|---|---|---|---|---|
| Full attention | $O(n)$ | $O(n^2)$ | $O(n)$ | 284M | 16.1 s | 2.3 FPS |
| Perceiver Attention | $O(1)$ | $O(n^2)$ | $O(1)$ | 287M | 16.8 s | 34.4 FPS |
| Ours | $O(1)$ | $O(n)$ | $O(1)$ | 312M | 1.4 s | 38.7 FPS |

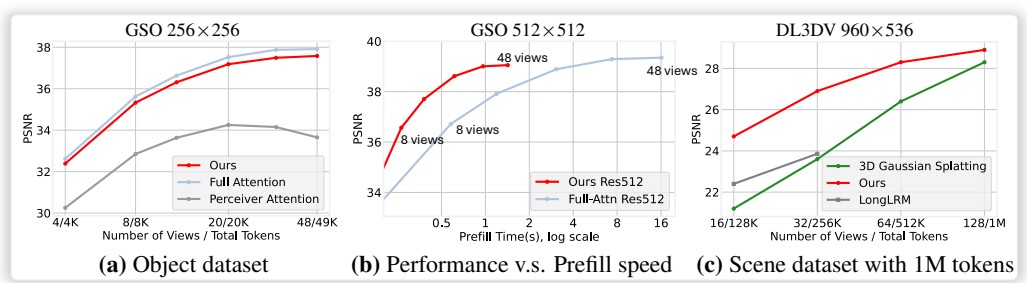

(a) Object dataset     (b) Performance v.s. Prefill speed     (c) Scene dataset with 1M tokens

Figure 4: **(a, b)** our method achieves quality comparable to full-attention models with significantly lower prefill latency, and it clearly outperforms perceiver-attention baselines. **(c)** On the high resolution scene dataset, our approach surpasses LongLRM, limited to 32 views, and outperforms 3D Gaussian Splatting with sparse views, remaining competitive up to 128 input views (1M total tokens).

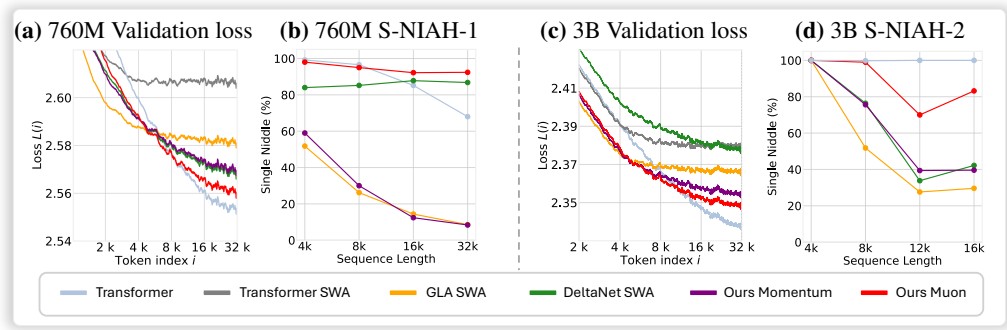

Figure 5: Language Model results. **(a, c)** Our model achieves lower per-position loss at larger token indices compared to GLA and DeltaNet at both 760M and 3B scale, indicating stronger long-context modeling capability. **(b, d)** Our model consistently outperforms GLA and DeltaNet in retrieval accuracy. Furthermore, our Muon variant consistently outperforms our Momentum variant.

## 6.2 LANGUAGE MODELING

**Datasets & Metrics.** We train our models on the Long-Data-Collections dataset (AI, 2024), using approximately 60B tokens. For evaluation, we employ the per-token loss metric from (Xiong et al., 2023; Lin et al., 2025), assessing models' ability to effectively use the full context. A monotonically decreasing loss indicates successful context utilization, whereas plateauing suggests limited context usage. Additionally, we report retrieval accuracy (Hsieh et al., 2024) at various sequence lengths.

**Model & Training details.** We integrate the sliding window-attention(SWA) layer directly into the Large-Chunk TTT layer by sharing Q, K, and V vectors with the fast-weight network, following GAU Hua et al. (2022a). The pseudocode for this design is in Algorithm 2. We trained models at two scales using a 32768-token sequence length: a 760M-parameter model with a 2048-token sliding window and a 3B-parameter model with a 4096-token sliding window. See more details in App. E.2.

**Baselines.** We compare against transformer, Gated Linear Attention (GLA) (Yang et al., 2024a), DeltaNet (Schlag et al., 2021; Yang et al., 2024c). To ensure fairness, we enhance both GLA and DeltaNet with the same SWA. Tab. 6 summarize the mechanism and training throughput of all methods. Fig. 5 presents experimental results and analysis. See more results and details in App. E.2.

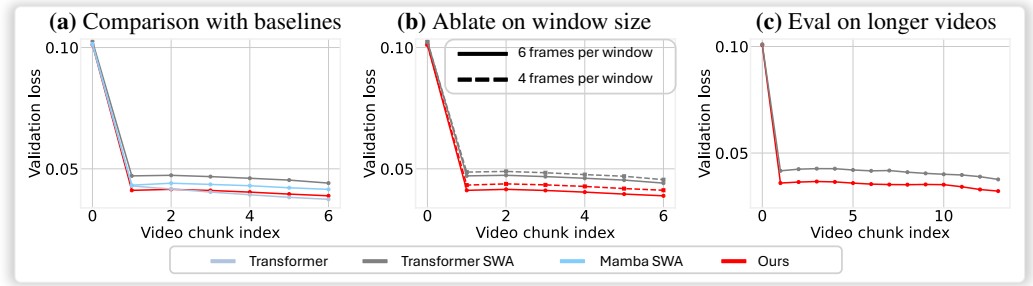

Figure 6: (a) We achieve comparable validation loss to the full-attention baseline and outperform both Mamba with sliding window and sliding window attention baselines. This improvement over SWA is consistent across different window sizes (b) and when evaluating on longer videos (c).

## 6.3 AUTOREGRESSIVE VIDEO DIFFUSION

We fine-tune the pretrained Wan 2.1 (Wang et al., 2025a) text-to-video diffusion model into an autoregressive video diffusion model. Specifically, we replace all bidirectional attention layers with our LaCT layers combined with sliding window attention, then fine-tune the model using an internal proprietary collection of videos, each accompanied by a short text prompt.

**Training details.** We train on 5-second videos at 16 FPS and $480 \times 832$ resolution, autoregressively denoising in 3 latent-frame chunks. Later we fine-tune the 1.3 billion parameter model with 10 second videos and 14 billion parameter model with 8.8 second videos. Each 8.8-second clip contains 56,160 visual tokens, resulting in interleaved noisy-clean chunks totaling 107K tokens under teacher-forcing training. Full details are listed in App. E.3.

**Baselines.** We compare our method against three baselines: sliding window attention (SWA) alone, Mamba2 (Dao & Gu, 2024) combined with SWA (using a similar in-layer hybrid strategy as our method), and full block-wise causal attention.

**Evaluation.** We evaluate all models on a collection of 2,000 videos after 5,000 training iterations by computing the denoising loss at five timesteps (550, 650, 750, 850, 950). Figure 6 plots the chunk-wise denoising loss across evaluated video frames. See Appendix E.3 for results on VBench (Huang et al., 2024). See the anonymous website in abstract for our autoregressively generated videos.

## 7 CONCLUSION

We presented LaCT, a novel model architecture that integrates large-chunk test-time training for capturing long context with window attention for modeling local structure. We validated LaCT across three diverse tasks spanning different modalities—novel view synthesis, language modeling, and autoregressive video diffusion—and demonstrate its effectiveness by achieving superior or competitive performance when compared to state-of-the-art baselines. LaCT achieves high GPU efficiency even with native PyTorch implementation with dozens of lines of code and supports efficient scaling up of the state size and more flexible designs in test-time training models and optimizers. By open-sourcing the code and weights, we hope that LaCT can advocate future research explorations into more performant architectures for long-context modeling.

**Reproducibility statement.** We provide pseudocode for large-chunk test-time training (Algorithm 1), the in-layer hybrid strategy with sliding-window attention (Algorithm 2), and context/tensor-parallel implementations (Algorithm 3, Algorithm 4). We provided the implementation details and hyperparameters in Appendix E for our main results. The source code, included in the supplementary materials, contains the model and full training details. It also provides a benchmarking code to evaluate the running speed of our method against two baselines.

**Acknowledgment.** We thank Ziwen Chen for processing the DL3DV dataset and providing the K-means clustering. We thank Nathan Carr, Feng Liu, Jianming Zhang, and Hailin Jin for their generous support on this project. We thank Haian Jin and Zexiang Xu for leading the LVSM project. We thank Baqiao Liu and Haoran Cai for the video data loader. We thank Guo Han for details on the language dataset. We thank Jeremy Bernstein for discussions on Muon.

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

**Below is the Appendix of the submission.**

## A  ANALYSIS ON DESIGN CHOICES

In this section, we analyze several key design choices in our model, focusing on both the novel view synthesis and language modeling tasks, where good metrics exist. Specifically, we evaluate the impact of state size (Fig. 7a), test-time optimizers (Fig. 7b), linear versus nonlinear fast weights (Fig. 8a), and per-token recurrence versus chunk-wise recurrence (Fig. 8b). Overall, we find that a large state size, advanced optimizers such as Muon, and nonlinear fast weights significantly improve our model's performance. For comparing chunk recurrence with per-token recurrence, in a controlled NVS experiment, our linear large-chunk recurrence strategy outperforms linear per-token recurrence with the same state size. For language modeling, where chunk structures are not inherent, our linear large-chunk recurrence variant—while initially underperforming per-token methods like GLA and DeltaNet—surpasses them when combined with a large nonlinear state and the Muon optimizer. We refer the readers to each figure and its caption for more detailed analysis.

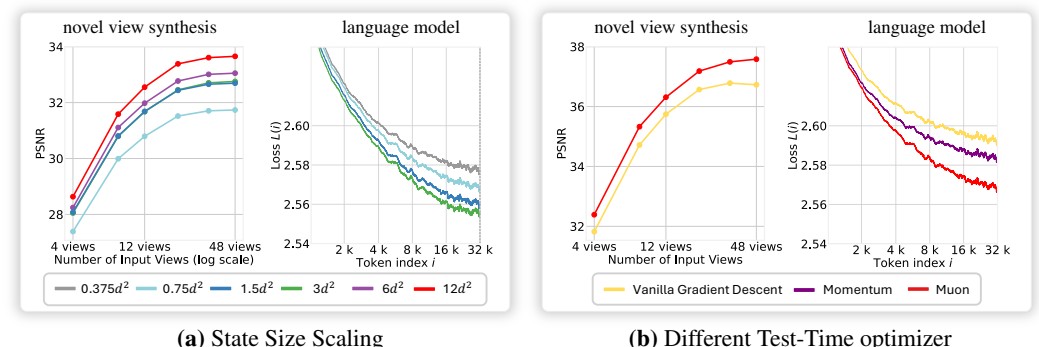

(a) State Size Scaling                (b) Different Test-Time optimizer

Figure 7: **(a)** Scaling up the state size consistently improves performance in both novel view synthesis and language modeling tasks. Note, the largest version has state size of $12d^2$ per block, totaling 40% of model weights as fast weights. **(b)** Comparison of test-time optimizers demonstrates Muon's surprising effectiveness over Vanilla Gradient Descent and Momentum.

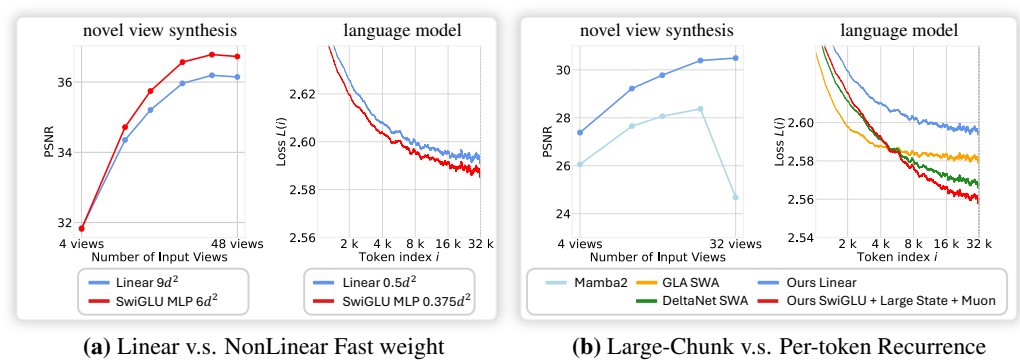

(a) Linear v.s. NonLinear Fast weight        (b) Large-Chunk v.s. Per-token Recurrence

Figure 8: **(a)** Nonlinear fast weights consistently outperform linear fast weights despite using smaller state sizes. **(b)** Our linear large-chunk recurrence approach significantly outperforms linear per-token recurrence (bidirectional Mamba2) for view synthesis tasks at the same state sizes. In language tasks, linear large-chunk recurrence of the same state size underperforms the GLA baseline, but when combined with larger nonlinear states and Muon test-time optimizer, it surpasses all per-token recurrence methods.

**State size scaling.**  These controlled experiments utilize a SwiGLU MLP for fast weights and the Muon as the test-time optimizer. For NVS, experiments were conducted on the object dataset. All models were trained for 167B tokens, using 14 stacked blocks and a model dimension $d = 768$.

To change the state size, we keep the head dimension fixed as model dimension. i.e. single head, and vary the intermediate dimension of SwiGLU MLP, such that the intermediate dimension ranges from 192 to 3072. The largest configuration results in a state size per model block as $12d^2$, totaling $40\%$ of model weights as fast weights. For the language model experiments, we use the 760 milion parameter setup, where the chunk size and sliding window attention (SWA) window size were set to 2048 tokens. We keep the intermediate dimension of the fast weight SwiGLU MLP the same as the head dimension. We increase the state size while proportionally decreasing the number of heads to maintain a fixed model dimension. Figure 7(a) demonstrates that larger state sizes consistently improve performance. Notably, the performance gap between small and large state sizes widens with increasing sequence length.

**Test-Time optimizer comparison.** We compare Muon with vanilla Gradient Descent (GD) and GD with momentum. Details on momentum implementation are in Appendix C. For NVS, we train all compared approaches for 671 tokens using model specs of 24 stacked blocks with model dimension of 768. Language modeling experiments used the 760M parameter setup. Figure 7(b) shows Muon consistently outperforming other optimizers.

**Linear v.s. NonLinear fast weight.** Our default fast weight function is a SwiGLU MLP without bias terms (nonlinear). We compare this against a simple linear fast weight, $f_W(x) = Wx$. Both are updated using the same online dot product loss for key-value association. Figure 8 (a) presents this comparison for NVS and language modeling. Although the linear fast weights were configured with a larger state size than the nonlinear SwiGLU, they achieved lower performance. NVS models were trained for 671B tokens with 24 blocks and $d = 768$. Language modeling used the 760M parameter setup.

**Large-chunk v.s. Per-token recurrence.** Figure 8(b) presents controlled experiments comparing our large-chunk recurrence with per-token recurrence. In the novel view synthesis (NVS) task, "Our Linear" variant employs a linear fast weight: $f_W(x) = Wx$ and is benchmarked against a Mamba-2 baseline (a linear per-token recurrence model) with an identical state size. To accommodate the bidirectional context required by NVS over input image tokens, the Mamba-2 baseline uses two Mamba-2 layers applied in opposite directions within each model block. Both our linear variant and this bidirectional Mamba-2 have state size of $d^2$ per block. Both of these two approaches employs a per-image window attention within each model block. Under this fair comparison, our linear large-chunk recurrence achieves significantly better view synthesis performance.

For the language modeling experiments also shown in Figure 8(b), the blue line "Our Linear" variant uses the same state size $(0.25d^2)$ as the GLA SWA baseline. It initially underperforms GLA SWA (blue line underperforms yellow line), likely because language data lacks the inherent chunk structures that benefit our basic linear chunk recurrence. However, when LaCT is equipped with a larger non-linear state $(1.5d^2)$ and Muon updates, we significantly outperform these per-token recurrence baselines.

## B  ADDITIONAL RELATED WORK

**Novel view synthesis.** Novel view synthesis (NVS) is a long-standing task at the intersection of computer vision, graphics, and computational photography, requiring algorithms to render images of a static scene from previously unobserved viewpoints. Optimization-based approaches, such as NeRF (Mildenhall et al., 2021) and 3D Gaussian Splatting (Kerbl et al., 2023), have achieved significant breakthroughs. These methods optimize a set of parameterized graphics primitives (i.e., explicit or implicit representations of radiance fields) through differentiable volumetric rendering to minimize reconstruction loss on input images. After an optimization process typically lasting tens of minutes, these approaches can render novel views photorealistically, and the optimized parameters form a 3D representation of the input scene.

Recently, data-driven approaches (Zhang et al., 2024; Jin et al., 2024a; Ziwen et al., 2024; Han et al., 2024; Liu et al., 2023) have also shown promising results. These methods can either directly render novel views or predict 3D representations given input images. Although successful on simpler object datasets, these methods often struggle with densely sampled scenes (e.g., scenes with over 100 input images). Our experiments demonstrate that our large-chunk test-time training approach outperforms

or achieves comparable performance to 3D Gaussian Splatting on challenging scene datasets with up to 128 input images with $960 \times 536$ resolution at challenging scene datasets.We hope our method will inspire further research into effectively scaling data-driven NVS methods to longer and more complex input sequences.

**Autoregressive video diffusion.** Current state-of-the-art video generation is dominated by bidirectional diffusion transformers operating in latent space (Brooks et al., 2024; Yang et al., 2024d; Polyak et al., 2024; Wang et al., 2025a). These methods factorize the video distribution into a sequence of conditional distributions based on noise levels, following diffusion processes (Sohl-Dickstein et al., 2015; Song et al., 2020) or flow matching (Lipman et al., 2022), then use a diffusion transformer to jointly learn all the conditional distribution. Autoregressive video diffusion (Alonso et al., 2024; Jin et al., 2024b; Valevski et al., 2024; Ruhe et al., 2024; Yin et al., 2024; Song et al., 2025) introduces an additional temporal dimension to this factorization, where the neural networks learns to model the conditional probability of the next chunks of videos at different noise levels, conditional on previous videos and noisier version of current video frames.

During training, some autoregressive methods employ teacher forcing, supervising the model on noisy video frames given previous clean context frames as condition (Alonso et al., 2024; Jin et al., 2024b; Valevski et al., 2024), though this can lead to low token utilization, i.e. only a small portion of tokens get supervision. To improve token efficiency, other techniques such as progressive noise injection (Ruhe et al., 2024) or the use of frame-independent noises (sometimes in a diffusion-forcing style) (Yin et al., 2024; Chen et al., 2024a; Sand-AI, 2025) have been proposed. When applying our large-chunk design to autoregressive video generation, we format the input sequence with interleaved clean and noisy chunks (see Equation 11). This strategy achieves over 50% token utilization and integrates effectively with our large-chunk TTT implementation, by only changing a few lines to constrain fast-weights are only updated on clean frame chunks.

## C LaCT Model Implementation Details

**State Size calculation.** Motivated by recent progress in LLM, we adopt SwiGLU-MLP (Shazeer, 2020) without bias terms as the fast-weight network. Our fast weights consists of three weight matrix $W = \{W_1, W_2, W_3\}$, and the forward pass of the fast weight model is:

$$f_W(x) = W_2 \left[ \text{SiLU}(W_1 x) \circ (W_3 x) \right] \tag{12}$$

where $\circ$ is an elementwise multiplication. We define $hd$ as the head dimension, $nh$ as the number of heads, and the intermediate dimension of the SwiGLU-MLP as $hd \times r$, where $r$ is a scaling multiplier.When $r > 1$, the intermediate dimension surpasses the input head dimension, which is the current common practice in LLMs. Thus, matrices $W_1, W_2 \in \mathbb{R}^{hd \times hd}$ and $W_3 \in \mathbb{R}^{hd \times hd \times r}$. Consequently, the total state size becomes $nh \times hd \times hd * r$. Given that typically the total head dimension across all heads equals the model dimension $d$ (i.e., $nh \times hd = d$), the total state size simplifies to:

$$\text{State Size} = \frac{d^2}{nh} * r. \tag{13}$$

Therefore, we can increase the state size either by reducing the number of heads or by increasing the intermediate dimension multiplier.

**FLOPs calculation.** When using then negative dot product loss as the online test-time training objectives, we don't need to compute the final results of $f_W(v)$. We only need to compute $W_1 v, W_3 v$ when running forward pass with keys $k$, thus there are two matmuls in the forward pass with keys. When computing the gradients, there are four matmuls. And in the final forward pass the queries, there would be three matmuls. So the total FLOPs with $n$ tokens would be:

$$\text{FLOPs} = 4n\frac{d^2}{nh}r + 8n\frac{d^2}{nh}r + 6n\frac{d^2}{nh}r = 18n\frac{d^2}{nh}r = 6 * \text{State Size} \tag{14}$$

**Model initializations.** We randomly initialize the linear layers using a standard deviation of 0.02. For the learnable initial fast weights, we initialize them with a standard deviation of $1.0/\sqrt{\text{fan-in}}$. Specifically, in the SwiGLU FFN fast weights, the matrices $w_1$ and $w_3$ have their fan-in set as the head

**Algorithm 1** Large Chunk Test-Time Training Layer Pseudocode

```python
def apply_fw(fast_weight, q):
    w1, w2, w3 = fast_weight
    hidden = silu(matmul(q, w1)) * matmul(q, w3) # [b, l, dh] = [b, l, d] x [b, d, dh]
    return matmul(hidden, w2)

def update(fast_weight, k, v, lr, use_muon=True):
    """
    Fast-weight update for a SwiGLU MLP using chunk of tensors.
    Args:
        fast_weight : tuple(w1, w2, w3) with shapes: w1, w3: [b, d, dh]; w2: [b, dh, d]
        k, v : key / value tensor of shape [b, l, d]
        lr: : per-token learaning rates of shape [b, l, 3] -> (lr1, lr2, lr3)
        use_muon : weather to apply Muon to orthogonalize the update
    Note:
        The head dimension for input tensors k, v, lr are assumed to be merged into the
            batch dimension. This simplifies shape annotation in this pseudocode.
    """
    # Forward with k:
    gate_before_act = matmul(k, w1) # [b, l, dh] = [b, l, d] x [b, d, dh]
    hidden_before_gate = matmul(k, w3) # [b, l, dh] = [b, l, d] x [b, d, dh]
    hidden = silu(gate_before_act) * hidden_before_gate

    # Backward:
    dhidden = matmul(v, w2.transpose(-1, -2)) # [b, l, dh] = [b, l, d] x [b, d, dh]
    dhidden_before_gate = dhidden * silu(gate_before_act)
    dgate = dhidden * hidden_before_gate
    dgate_before_act = silu_backprop(dgate, gate_before_act)

    # Compute gradients:
    w2.grad = -matmul(hidden.transpose(-1, -2), v * lr2) # [b, dh, d]
    # [b, d, dh] = [b, d, l] x [b, l, dh]
    w1.grad = -matmul((k * lr1).transpose(-1, -2), dgate_before_act)
    w3.grad = -matmul((k * lr3).transpose(-1, -2), dhidden_before_gate)

    # Weight update
    if use_muon:
        for w in fast_weight:
            w.grad = zeropower_via_newtonschulz5(w.grad)
    for w in fast_weight:
        w = (w - w.grad) / (w - w.grad).norm(dim=1) * w.norm(dim=1)
    return fast_weight

def silu_backprop(dy, x):
    sigma = sigmoid(x)
    return dy * sigma * (1 + x * (1 - sigma))

############################ MultiHead LaCT layer ############################
# x: input sequence [b, l, d], b is the batch dim, l is length, d is model dimension
# fast_weight: tuple of initial fast weights-(w1, w2, w3); w1, w3 of shape [nh, d, dh
    ], w2: [nh, dh, d]
# ttt_config: list of (operation, begin, end) tuples
qkv = silu(LinearQKV(x)) # [b, l, d * 3]
q, k, v = rearrange(qkv, `b l (nh hd) -> (b nh) l hd`, nh=num_heads).split(3)
q, k = q / q.norm(-1), k / k.norm(-1)
lr = softplus(LinearLR(x) + const_lr_bias) # [b, l, 3 * num_heads]
lr = rearrange(lr, `b l (nh 3) -> (b nh) l 3`, nh=num_heads)
fast_weight = repeat(fast_weight, dim=0, repeat=b) # [nh, ...] -> [b * nh, ...]

o = zeros_like(v) # [b * nh, l, hd]
for mode, begin, end in ttt_config:
  qi, ki, vi, lri = q[:, begin:end], k[:, begin:end], v[:, begin:end], lr[:, begin:
      end]

  if mode == 'update_then_apply': # figure-3(a, b) bidirectional attention
    fast_weight = update(fast_weight, ki, vi, lri, use_muon)
    o[:, begin: end] = apply_fw(fast_weight, qi)

  elif mode == 'apply_then_update': # figure-3(b) shifted block-wise causal
    o[:, begin: end] = apply_fw(fast_weight, qi)
    fast_weight = update(fast_weight, ki, vi, lri, use_muon)

  elif mode == 'update_only':
    fast_weight = update(fast_weight, ki, vi, lri, use_muon)

  elif mode == 'apply_only':
    o[:, begin: end] = apply_fw(fast_weight, qi)

o = RMSNorm(o) # per-head norm
o = LinearOutput(rearrange(o, `(b nh) l hd -> b l (nh hd)`, nh=num_heads))
return o
```

---

**Algorithm 2** LaCT Layer with In-Layer Hybrid Window Attention Pseudocode

---

```
# Input:
# x: input sequence [b, l, d], b is the batch dim, l is length, d is model dimension
# fast_weight: tuple of initial fast weights-(w1, w2, w3); w1, w3 of shape [d, dh],
    w2 of shape [dh, d]
# ttt_config: list of (operation, begin, end) tuples

q, k, v = LinearQKV(x).split(3)

#### Local quadratic-cost window attention
attn_q = q * learnable_q_scale + learnable_q_offset # per-channel rescale and shift
attn_k = k * learnable_k_scale + learnable_k_offset # per-channel rescale and shift
attn_o = local_softmax_multihead_attn(attn_q, attn_k, v, attn_mask)

#### large chunk test-time training for long memory
q, k = rearrange(q, k, `b l (nh hd) -> (b nh) l hd`, nh=num_heads)
q, k = silu(q), silu(k)
q, k = q / q.norm(-1), k / k.norm(-1)
lr = softplus(LinearLR(x)) # [b, l, 3 * num_heads]
lr = rearrange(lr, `b l (nh 3) -> (b nh) l 3`, nh=num_heads)

# Perform update and apply_fw operations iteratively over chunks of tokens.
lact_o = lact(fast_weight, q, k, v, lr, ttt_config)
lact_o = RMSNorm(lact_o)

scale_per_head = rearrange(silu(Linear(x)), `b l nh -> (b nh) l 1`, nh=num_heads)
lact_o = lact_o * scale_per_head
lact_o = rearrange(lact_o, `(b nh) l hd -> b l (nh hd)`, nh=num_heads)

#### Merge attention results (shape: [b, l, d])
o = attn_o + lact_o

o = LinearOutput(o)

return o
```

---

dimension, while the fan-in of $w_2$ is the intermediate dimension of the SwiGLU FFN fast weights. Additionally, when local window attention is incorporated within the LaCT layer, we introduce four extra learnable embeddings: two scales and two reshifts for queries and values. We initialize the scale embeddings as ones and the reshift embeddings as zeros.

**Details of Muon.** Muon (Jordan et al., 2024) is a recently proposed optimizer that orthogonalizes the matrix gradients during updates of matrix weights. It utilizes Newton-Schulz iterations to achieve orthogonalization. Given a matrix gradient $G$, Muon first normalizes it as $G_0 = G/|G|F$, then iteratively applies:

$$\mathbf{G}_k = a\mathbf{G}_{k-1} + b(\mathbf{G}_{k-1}\mathbf{G}_{k-1}^{\mathrm{T}})\mathbf{G}_{k-1} + c(\mathbf{G}_{k-1}\mathbf{G}_{k-1}^{\mathrm{T}})^2\mathbf{G}_{k-1}, \tag{15}$$

where the constants $a, b, c$ are carefully chosen for optimal convergence. Following the original implementation, we set $a = 3.4445$, $b = -4.7750$, $c = 2.0315$, and perform five iterations.

Each Muon iteration requires three matrix multiplications, resulting in a computational cost per fast weight head of $2hd^3r + 2hd^3 + 2hd^3r = hd^3(4r + 2)$ FLOPS. Hence, the total computation for five iterations across all fast weights is:

$$5 \times nh \times hd^3 \times (4r + 2). \tag{16}$$

For the case where $r = 1$ (head and intermediate dimensions are equal), the total computational cost simplifies to:

$$30 \times nh \times hd^3 = 30 \times hd \times \text{State Size}. \tag{17}$$

This indicates that the computational overhead of Muon becomes less significant than computing token outputs only if the online chunk size exceeds $\frac{5}{3}hd$.

**Rotation invariance.** Softmax attention and linear attention exhibit rotation invariance: rotating the queries and keys by the same rotation matrix does not alter the output. This property is also used in developing relative positional encodings, like RoPE (Su et al., 2023). In contrast, our SwiGLU and Linear Fast Weight components do not possess this property.

**Implementing momentum for test-time optimizers.** Muon uses momentum by default. Following Titans (Behrouz et al., 2024), we implement momentum in the test-time optimizer by predicting a scalar momentum $\beta_i$ from each token:

$$\beta_i = \sigma(\text{Linear}(\boldsymbol{x_i})), \tag{18}$$

where $\sigma$ is the sigmoid function. This $\beta_i$ is then averaged over all tokens in the chunk, and the average momentum is applied as follows:

$$
\begin{aligned}
g &\leftarrow \sum_i^b \eta_i \nabla_W \mathcal{L}(f_W(k_i), v_i), \\
M &\leftarrow M(\sum_i^b \beta_i/b) + g, \\
W &\leftarrow \text{weight-update}(W, M),
\end{aligned}
\tag{19}
$$

where the weight-update can be simple subtraction followed by L2 normalization normalization (as in Equation 8). or Muon update before subtraction (as in Equation 9).

## C.1 PSEUDOCODE

See Algorithm 1 for pseudocode of a full LaCT layer. For details on how to mix local window attention inside each layer with shared query, value, embedding, see Algorithm 2 for pseudocode.

# D LACT ARCHITECTURE DETAILS FOR N-DIMENSIONAL DATA

## D.1 LACT ARCHITECTURE FOR NOVEL VIEW SYNTHESIS

Novel view synthesis (NVS) renders images of a static scene from novel viewpoints. Formally, given a set of $N$ input posed images $\{(I_i, P_i)\}_{i=1}^N$ of a static scene, where $I_i \in \mathbb{R}^{H \times W \times 3}$ is an RGB image and $P_i$ is its corresponding camera pose, the model needs to synthesize new images from novel camera poses $P_{\text{novel}}$ that typically do not overlap with the input views.

Traditional methods in 3D vision usually solved the NVS task by reconstruction and rendering. The reconstruction compresses the posed input into a compact representation and then the render renders the novel view from it. Our method mimics such pipeline, where we first compress the posed input images into fast weights by the 'Update Operation' (Sec 4.1) in LaCT. Then we render the novel view images from the information in the compressed weights with the 'Applying Operation'.

In details, we first convert the input and output into tokens. The camera pose $P$ for each view is represented in dense ray information for each pixel (usually from the camera intrinsics and extrinsic), i.e., $P = (\text{rays}_o, \text{rays}_d)$. $\text{rays}_o \in \mathbb{R}^{H \times W \times 3}$ is the 3D coordinate for the origins of the ray, and $\text{rays}_d \in \mathbb{R}^{H \times W \times 3}$ is the direction of the ray. We follow GS-LRM (Zhang et al., 2024) to use the Plücker ray embedding for the rays. Plücker ray embedding computes the cross product between the ray origin and ray direction for a normalization. The final positional embedding is a concatenation of the ray's origin, the ray's direction, and the cross product of the above two: $[\text{rays}_o, \text{rays}_d, \text{rays}_o \times \text{rays}_d]$. We add ray's origin into the embedding since different origins in the same ray can results in different colors due to the occlusions. We then use patchifying and two different Linear layers to convert the RGB map and ray map (i.e., the positional embedding) into tokens. For the posed RGB input images, we simply the sum the RGB embedding and pose embedding as model input. For the novel view cameras, we only use the pose embedding.

We illustrate the design of NVS's LaCT block in Fig. 9. We first apply the attention for each image (either input or the novel target). The attention is bidirectional for the tokens belonging to the same image, and is independent among different images. Then, the TTT update operation is applied to all input tokens, i.e., all tokens that belonging to all input images. The updated weight then is applied to all tokens. The two updates blocks in Fig. 9 take the same updated fast weight thus can be combined, and we left two 'Apply' block for clearance. Note that the original NVS task definition renders novel views independently. We here supervise multiple novel image poses and their corresponding images in a single data point for better training efficiency. Given the design, the novel images are

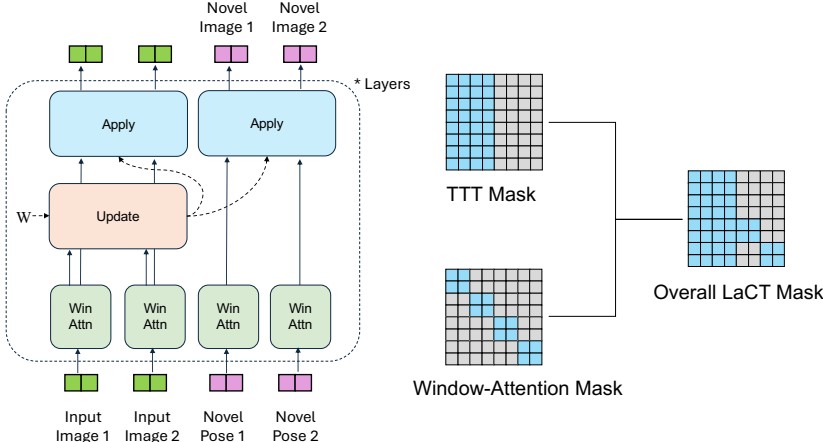

Figure 9: Detailed LaCT model for our Novel-View Synthesis. Dashed line indicates flow of fast weight. Solid line indicates flow of tokens. Window attention is bidirectional within a single image, either the input image or the novel target image. TTT updates over all input tokens and apply to all tokens.

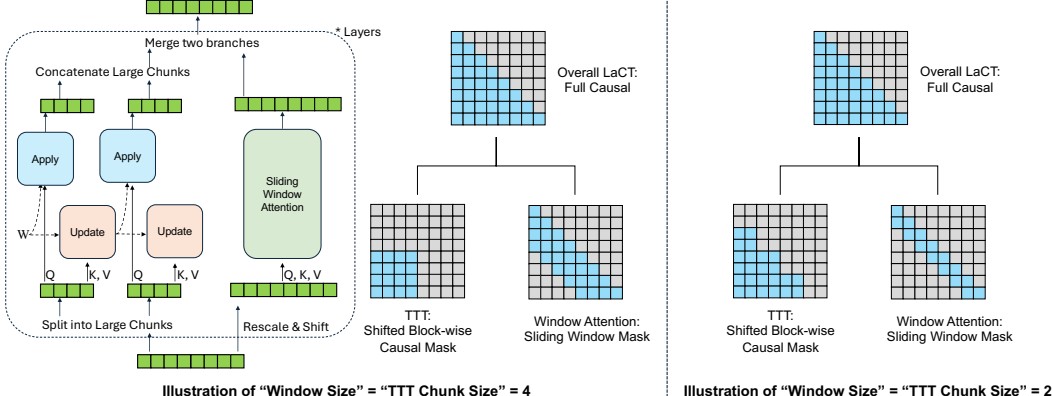

Figure 10: Detailed LaCT model for language models. Dashed line indicates flow of fast weight. Solid line indicates flow of tokens. We illustrate with TTT chunk size 4 or 2, and the actual chunk size is over 2048 in LaCT. We take the parallel design for the window attention and TTT block with shared QKV. The overall mask is the causal mask.

independent to each other, which is illustrated in the 'Overall LaCT Mask' in the right of Fig. 9. The layer normalization layer, the residual connections, and the feed-forward network is omitted for clarity. The block is repeated by number of layers times to formulate the full model. The general model largely follow the design of the encoder-decoder model in LVSM (Jin et al., 2024a), except we use TTT in replace of transformer for long-context modeling.

For actually using this model for NVS task during inference, we first get the updated fast weight with all input images. Then, we would not change the fast weight during the rendering process (i.e., the process to convert novel camera poses to the novel images). The LaCT during rendering would be similar to a ViT (Vision transformer) architecture despite having two feed-forward networks: the feed-forward network from the fast weight stores the scene information, and the feed-forward network from the slow weight (i.e., the FFN in Fig. 2) stored the world knowledge like physical rendering rules.

## D.2 LaCT ARCHITECTURE FOR LANGUAGE MODELS

Autoregressive Language Models (LM) predicts the distribution of the next tokens $p_\theta(x_n|x_1, \ldots, x_{n-1})$ from its history context. It is a factorization of the full sequence distribu-

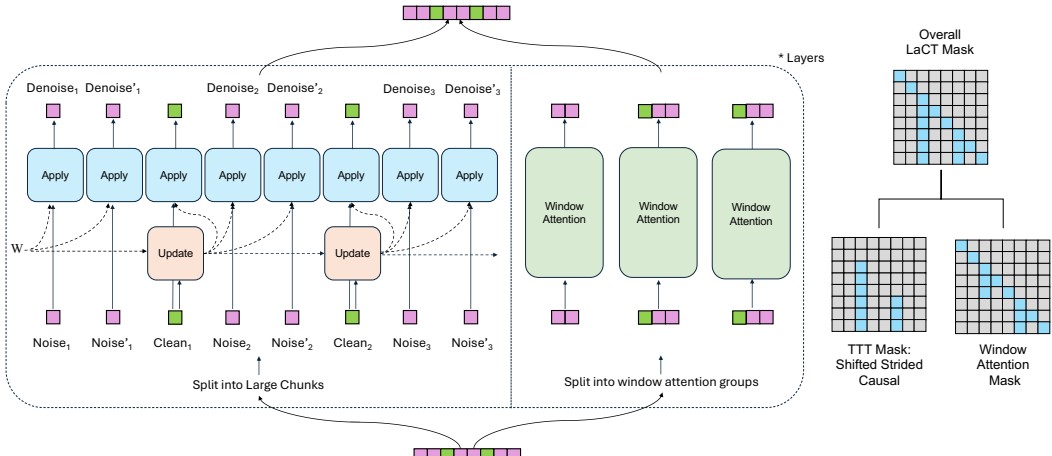

Figure 11: Detailed LaCT model for autoregressive video generation by diffusion model. The purple tokens are noisy frame chunk for training the diffusion model. The green tokens are clean frame chunk. Each token is a large chunk in TTT, e.g., 3 consecutive frames with 4680 tokens in total. Noise and Noise' are two noisy frames with independent Gaussian noise and time stamps over the same clean frame. We denoise them simultaneously to improve the utilization. Dashed line indicates flow of fast weight. Solid line indicates flow of tokens. We take the parallel design for the window attention and TTT block with shared QKV. The TTT mask is shifted (i.e., started with apply) strided causal. The window attention excludes the attention from clean frame to future noisy frame, and also excludes the attention between the independent noisy frames.

tion $p_\theta(x_1, \ldots, x_n)$ through chain rule $p_\theta(x_1, \ldots, x_n) = p_\theta(x_1)p_\theta(x_2|x_1) \ldots p_\theta(x_n|x_1, \ldots, x_{n-1})$. Thus it requires a token-level causal mask (demonstrated in the topright of Fig. 10) and this is the main difficulty for the large-chunk design in LaCT. We use a combination of TTT layer with 'Shifted Causal Block Mask' (introduced before in Fig. 3c) and a sliding window attention to facilitate it. By shifting the mask of TTT, it excludes the information leakage from future tokens. As shown in the right part of Fig. 10, the overall dependency mask is the union of the TTT mask and the sliding window attention mask. To achieve a token-level causal mask without bubbles, the only requirement is that the window size of the sliding window attention is greater or equal to the chunk size from the TTT. We illustrate two example of such mask with 'Window Size' = 'TTT Chunk Size' = 2 or 4. The actual chunk size and attention window size is above 2048 in our implementation for better utilization and state size scaling (discussed in Sec. 3.2).

As illustrated in the left most of Fig. 10, we employ a parallel design of the TTT layer and sliding window attention to save the number of model parameters and computation FLOPs. In details, the query (Q), key (K) and value (V) are shared between the TTT layer and window attention. Sliding window attention is an attention with constant window size over the past history, starting from the target tokens. For the TTT layer, we start with an apply operation over the first chunk using the initialized fast weight (i.e., unupdated yet). The 'apply' operation is followed by the 'update' operation over the first chunk. In this way, the 'apply' operation would not see information inside the current chunk to avoid leaking future token information inside the chunk. Alternatively using 'apply' followed by 'update' over subsequent blocks completes the desired 'shifted block-wise causal mask' illustrated in Fig. 10. For details of the parallel design, please refer to the Pseudocode Algorithm 2.

We use the multi-head design for both TTT layer and window attention, although their number of heads are different. We empirically take less number of heads for TTT layer (i.e., larger head dimension) to enable larger state size, as state size is propotional to the head dimension in our design (Sec. C, and Equation 13). By default, we use four heads in langugae model experiments. For positional encoding, we use the same RoPE as the window attention branch.

### D.3 LaCT ARCHITECTURE FOR AUTOREGRESSIVE VIDEO DIFFUSION

Chunkwise autoregressive video diffusion generates videos by iteratively denoising sequential chunks of video frames, conditioned on previously generated clean frames. Each chunk can contain several video frames and span thousands of visual tokens. We use teacher-forcing training by interleaving

noisy and clean frame chunks. Specifically, a video of N frame chunks is structured as:

$$S = [X_1^{\text{noise}}, X_1, X_2^{\text{noise}}, X_2, \ldots, X_N^{\text{noise}}] \tag{20}$$

where each noisy chunk $X_i^{\text{noise}}$ is produced by adding unit Gaussian noise $\epsilon$ to the $i$-th clean video chunk as $X_i^{\text{noise}} = X_i(1 - t_i) + \epsilon t_i$ and $t_i \in [0, 1]$ denotes the strength of chunk-independent noise. However, compared to previous methods that employs progressive (Ruhe et al., 2024) or frame-independent noise strategies (Yin et al., 2024) like diffusion forcing (Chen et al., 2024a), our teacher-forcing formulation in Equation 20 only uses around $50\%$ of the tokens of the entire sequence to compute the denoising loss. To improve token utilization, we consider an alternate approach by repeating each video chunk with two noise levels in the training sequence as:

$$S = [X_1^{\text{noise}}, X_1^{\text{noise}*}, X_1, X_2^{\text{noise}}, X_2^{\text{noise}*}, X_2, \ldots, X_N^{\text{noise}}, X_N^{\text{noise}*}], \tag{21}$$

where $X_i^{\text{noise}}$ and $X_i^{\text{noise}*}$ represent two different noise levels applied to each clean video chunk $X_i$. This increases token utilization from $50\%$ to about $67\%$. While more repetition could further increase token utilization, it would also reduce training sample diversity; thus, we limit the repetition to twice. We use such repeating strategy when training the 1.3 billion parameter video diffusion model on five seconds videos.

To process these interleaved noisy and clean chunks, LaCT fast weights are updated exclusively using the clean video chunks. These updated weights are then applied to the current clean chunk and subsequent noisy chunks. The integrated local window attention uses a window size of two frame chunks and employs a block-wise causal mask. This mask allows noisy chunks to attend only to themselves and the immediately preceding clean chunk. By doing this, we main bidirectional dependies within each chunk and causal dependency across chunks.

Similarly to our language model experiments, we integrate the TTT and local window attention into the same layer. Figure 11 illustrates this design for autoregressive video generation. The input sequence depicted in the figure follows Equation 21 with each video chunk repeated twice with different noise levels. The pink color in the figure indicates noisy chunks and the green color indicates clean chunks.

# E  EXPERIMENTAL DETAILS

## E.1  NOVEL VIEW SYNTHESIS

**Datasets & metric.** We evaluate our approach on both object-level and scene-level datasets. We use Objaverse dataset (Deitke et al., 2023) for object-level training, and render 32 random views per object, following the setup from LVSM (Jin et al., 2024a) and GS-LRM (Zhang et al., 2024). After training, we perform evaluations on the Google Scanned Objects (GSO) dataset (Downs et al., 2022), at resolutions of $256 \times 256$ and $512 \times 512$. To ensure stablized evaluation results, we render at fixed view points instead of random view points as in training. Each evaluation involves 4–48 input views and 8 novel views per object. For scene-level evaluations, we adopt the challenging DL3DV scene dataset (Ling et al., 2024), which contains over 11K training scenes and 140 testing scenes, each with approximately 300 views. Evaluations are performed at a resolution of $960 \times 540$ [2]. We report Peak Signal-to-Noise Ratio (PSNR) in the paper's main figures, and other metrics Structural Similarity Index Measure (SSIM) and LPIPS (Zhang et al., 2018) can be found in Tables below. For DL3DV evaluation, we follow the original paper (Ling et al., 2024) and LongLRM (Ziwen et al., 2024) to use one frame from every 8 frame in the full video sequence as the target frames. The input frames are from the K-means clustering of all frames as in (Ziwen et al., 2024).

**Model details.** Our models consist of 24 stacked LaCT blocks, each with a model dimension of 768. The detail of such block is illustrated in Sec. E.1: Unless otherwise specified, we use a single-headed fast-weight SwiGLU-MLP with a hidden dimension of 1536. The window attention has 12 heads with head dimension 64, and is equipped with QK-normalization (Henry et al., 2020). The Feed-forward Network has 3072 as its intermediate hidden dimension. The model has a total of 312M parameters, of which 84M are fast weights (i.e., $6d^2$ per block). We use an fast-weight lr initialization of 0.01 by

---

[2]The original DL3DV 960p frames released in resolution of $960 \times 536$. To accommodate the patch-size 8 in our modeling, we crop it to $960 \times 536$ and the camera parameters are changed accordingly.

setting 'const_lr_bias' in Algorithm 1 to $\mathrm{softplus(const\_lr\_bias)} = 0.01$. As we used Muon in fast weight update for NVS, LaCT is not sensitive to lr scale as discussed in Sec. 4.2.

**Baselines.** For object-level evaluations, we compare against two baselines, including a full-attention model, and a register-attention model in a Perceiver style (Jaegle et al., 2021), In the full-attention baseline, we replace the TTT layer in our model with a block-wise causal attention layer, where the input tokens interact bidirectionally and the novel view tokens cross-attend to the input tokens. Such a design resembles our method's prefill and parallel decoding strategy described in Section 5.1, and the key-value caches of the input tokens server as scene representations for novel view renderings. In the Perceiver-style model, we replace half of the TTT layers with input-to-register full-attention layers and the remaining half with register-to-novel-view cross-attention layers. Such a model first compresses the input tokens into a constant set of register tokens and then decoding the novel view tokens by attending to the registers. For scene-level evaluations, we compare against a state-of-the-art long-sequence 3D reconstruction work LongLRM (Ziwen et al., 2024) that applies Mamba (Gu & Dao, 2023) hybrid with full attention to predict 3D Gaussian splats (Kerbl et al., 2023). We also include comparisons with pure optimization-based 3D Gaussian splatting methods. Tab. 2 compares the computational complexity of the baseline models and our models.

**Training details.** For object-level experiments, we first train all the model with 671B tokens at 8 input view and 8 novel view setting at a resolution of $256 \times 256$. We then finetune them with $512 \times 512$ resolution for an additional 587B tokens. For scene dataset, we first pre-train our model first with 32 input views and 32 novel views at $128 \times 128$ resolution for 1.5T tokens, then progressively finetune at larger resolutions, larger field-of-views, and more input views. The finetuning is always go with a non-squared FoV to match the raw data. Non-squared FoV has larger view range than the squared FoV, thus is harder. The input and novel views in fine-tuning are both 64 to support better view coverage. The curriculum of the fine-tuning resolution is set as $128 \times 72$, $256 \times 144$, $512 \times 288$, and $960 \times 536$. The training tokens for each stage is around 100B. High-resolution models (starting from $512 \times 288$) are trained with inner-chunk context parallelism (Sec. 4.4).

At each training stage, we always use AdamW with linear learning rate warmup and weight decay of $0.05$. The peak learning rate of the pre-training is $4e-4$. During fine-tuning, we use smaller learning rate (usually $1e-5$ to $5e-5$).

The training is completed with 64 A100 GPUs. The pre-training takes 8 days, and each fine-tuning stage is about 12hours (thus 2 days in total).

**Detailed Result Numbers** We here provided the detailed number for object-level results on the GSO dataset (at resolution $256 \times 256$ in Table 3, $512 \times 512$ in Table 4) and DL3DV evaluations (at resolution $960 \times 536$ in Table 5).

Table 3: 256-Res object-level novel view synthesis results on GSO. Both the input and output are with resolution $256 \times 256$. ↑: higher is better, ↓: lower is better.

| Input Views | # Input Tokens | LaCT | | | Full Attention | | | Perceiver Attention | | |
|---|---|---|---|---|---|---|---|---|---|---|
| | | PSNR (↑) | LPIPS (↓) | SSIM (↑) | PSNR (↑) | LPIPS (↓) | SSIM (↑) | PSNR (↑) | LPIPS (↓) | SSIM (↑) |
| 4 | 4,096 | 32.4 | 0.030 | 0.962 | 32.6 | 0.029 | 0.964 | 30.3 | 0.039 | 0.950 |
| 8 | 8,192 | 35.3 | 0.019 | 0.976 | 35.6 | 0.018 | 0.978 | 32.8 | 0.026 | 0.967 |
| 12 | 12,288 | 36.3 | 0.017 | 0.980 | 36.6 | 0.015 | 0.982 | 33.6 | 0.023 | 0.971 |
| 20 | 20,480 | 37.2 | 0.015 | 0.982 | 37.5 | 0.014 | 0.984 | 34.3 | 0.021 | 0.974 |
| 32 | 32,768 | 37.5 | 0.014 | 0.982 | 37.9 | 0.013 | 0.985 | 34.2 | 0.021 | 0.974 |
| 48 | 49,152 | 37.6 | 0.014 | 0.983 | 37.9 | 0.013 | 0.985 | 33.7 | 0.022 | 0.972 |

Table 4: 512-Res object-level novel view synthesis results on GSO. Both the input and output are with resolution $512 \times 512$ comparison across methods. ↑: higher is better, ↓: lower is better.

| Input Views | # Input Tokens | LaCT | | | Full Attention | | |
|---|---|---|---|---|---|---|---|
| | | PSNR (↑) | LPIPS (↓) | SSIM (↑) | PSNR (↑) | LPIPS (↓) | SSIM (↑) |
| 4 | 16,384 | 33.4 | 0.029 | 0.969 | 33.6 | 0.027 | 0.971 |
| 8 | 32,768 | 36.6 | 0.020 | 0.979 | 36.7 | 0.017 | 0.982 |
| 12 | 49,152 | 37.7 | 0.017 | 0.983 | 37.9 | 0.015 | 0.985 |
| 20 | 81,920 | 38.6 | 0.016 | 0.984 | 38.9 | 0.013 | 0.987 |
| 32 | 131,072 | 39.0 | 0.015 | 0.985 | 39.3 | 0.013 | 0.988 |
| 48 | 196,608 | 39.1 | 0.015 | 0.985 | 39.3 | 0.012 | 0.988 |

Table 5: 960P scene-level novel view synthesis results for LaCT on DL3DV. Both the input and output are with resolution $960 \times 536$ (width x height). ↑: higher is better, ↓: lower is better.

| Input Views | # Input Tokens | PSNR (↑) | LPIPS (↓) | SSIM (↑) |
|---|---|---|---|---|
| 16 | 128,640 | 24.7 | 0.224 | 0.793 |
| 32 | 257,280 | 26.9 | 0.185 | 0.837 |
| 64 | 515,520 | 28.3 | 0.169 | 0.857 |
| 128 | 1,031,520 | 28.9 | 0.166 | 0.861 |

### E.2 LANGUAGE MODELING

**Datasets & Metrics.** We train our models on the Long-Data-Collections dataset (AI, 2024), containing approximately 68.8B tokens tokenized using Mixtral tokenizer (32,000 codebook size). The dataset is a mix of 41.4% General Data (e.g., RedPajama-Book, RedPajama-ArXiv, 1B tokens from RedPajama, and a Pile subsample) and 58.6% Instruction Data (e.g., UL2 Oscar, NI, and P3). To evaluate long-context capabilities, we utilized the per-token loss metric from (Lin et al., 2025). A consistently decreasing per-token loss across the input sequence indicates effective use of the entire context, while a plateau suggests an inability to leverage information beyond that point. Specifically, we evaluated next-token prediction loss on 2.5B tokens from the Book-3 dataset (Gao et al., 2020) for our 760M-parameter model, and on 5B tokens for our 3B-parameter model. Additionally, we measured retrieval accuracy using the RULER benchmark (Hsieh et al., 2024) across various sequence lengths to assess context memorization and information retrieval, evaluating up to the trained sequence length.

**Model details.** We modified the original LaCT block by removing its window-attention layer. Instead, we incorporate a causal sliding window-attention(SWA) layer directly into the Large-Chunk TTT layer. The SWA layer shares the same Q, K, and V vectors as the fast-weight network, except that a per-channel leranable scale and shift is applied to Q and K before they are fed to the SWA layer (as done in GAU (Hua et al., 2022a)). We sum up the output of the SWA layer and that of the TTT layer, where the output of the TTT layer is scaled by another per-head learnable scalar. We use an fast-weight lr initialization of 0.001 by setting 'const_lr_bias' in Algorithm 1 to $\mathrm{softplus}(\text{const\_lr\_bias}) = 0.001$. We illustrate this archtecture in Figure 10.

To ensure a fair comparison with baselines in terms of trainable parameters.we adjusted the LaCT block's extra learnable initial fast weights $W = [W_1, W_2, W_3]$. o reduce parameters, we employed a low-rank version for $W_1, W3$ with a rank of 32. For instance, if $W_1 \in \mathcal{R}^{d \times d}$, its low-rank initial fast weight is $W_1 = L \cdot R + 0.5 * I_d$, where $L \in \mathcal{R}^{d \times 32}, R \in \mathcal{R}^{32 \times d}$, and $I_d$ is identity matrix. This reduces the extra trainable parameters for the fast weights in each block to $128 * \text{model-dim} + \frac{1}{\text{num-heads}}(\text{model-dim}^2)$. Additional minor parameters for learning rate projection, per-head scalers, an extra RMSNorm, and the SWA's learnable scale and shift are of order $O(\text{model-dim})$. Standard blocks typically have $12 \cdot \text{model-dim}^2$ parameters. Our approach adds approximately $\frac{1}{\text{num-heads}}(\text{model-dim}^2)$ extra parameters, which is less than 3% of total trainable weights with four heads, and below 4.5% with two heads. By default, LaCT use four heads in the experiments, unless noted otherwise, which means that the default state size per block is $\frac{3}{4}(\text{model-dim}^2)$.

**Baselines.** We compare our approach with full attention, Gated Linear Attention (GLA) (Yang et al., 2024a), DeltaNet (Schlag et al., 2021; Yang et al., 2024c). To ensure fairness, we enhance both GLA and DeltaNet with the same sliding window attention. As pointed out in previous work (Lin et al., 2025; Xiong et al., 2023; Men et al., 2024), a large RoPE (Su et al., 2023) base is critical for transformers in long-context training, thus we adopt a large RoPE base of 1 million for training with 32K token contexts whenever softmax attention is used. Tab. 6 compares the mechanism and computing complexity of the baseline methods and our method. Training throughput (tokens per second per GPU, TPS) was using a 3B-parameter model on eight A100-40GB SXM4 GPUs with activation checkpointing and FSDP. At the 3 billion parameter scale, all models use 24 softmax attention heads. The GLA baseline has eight linear attention heads with heads dimension as 384, resulting in a total state size of $384d$, with $d = 3072$ representing the model dimension. DeltaNet employs 24 linear attention heads, each with a dimension of 128, leading to a total state size of $128d$. Our approach uses four TTT heads with head dimension as 768, and since each block has three fast weights, the total state size is $2304d$.

Table 6: Comparison of baseline methods in terms of state size, training throughput (measured in tokens per second, TPS), update rules, and memory read-out mechanisms. Training throughput is evaluated using a 3B-parameter model with 32K-sequence length on A100-40GB GPUs.

| | State size | Train TPS | Update Rule | Memory read-out |
|---|---|---|---|---|
| Transformer | – | 4.1K | – | – |
| Transformer SWA | – | 6.4K | – | – |
| *Per-token recurrence* | | | | |
| GLA SWA | $384d$ | 5.0K | $\mathbf{S}_t \leftarrow \mathbf{S}_{t-1}\mathrm{Diag}(\boldsymbol{\alpha}_t) + \mathbf{v}_t\mathbf{k}_t^\top$ | $\mathbf{o}_t = \mathbf{S}_t\mathbf{q}_t$ |
| DeltaNet SWA | $128d$ | 5.1K | $\mathbf{S}_t \leftarrow \mathbf{S}_{t-1}(\mathbf{I} - \beta_t\mathbf{k}_t\mathbf{k}_t^\top) + \beta_t\mathbf{v}_t\mathbf{k}_t^\top$ | $\mathbf{o}_t = \mathbf{S}_t\mathbf{q}_t$ |
| *Large-chunk recurrence* | | | | |
| Ours GD | $2304d$ | 5.0K | $W \leftarrow \mathrm{L2norm}(W - \sum_i^b \eta_i \nabla_W \mathcal{L}_i)$ | $\mathbf{o}_t = f_W(\mathbf{q}_t)$ |
| Ours Momentum | $2304d$ | 4.9K | $M \leftarrow \beta M + \sum_i^b \eta_i \nabla_W \mathcal{L}_i;\ W \leftarrow \mathrm{L2norm}(W - M)$ | $\mathbf{o}_t = f_W(\mathbf{q}_t)$ |
| Ours Muon | $2304d$ | 4.3K | $M \leftarrow \beta M + \sum_i^b \eta_i \nabla_W \mathcal{L}_i;\ W \leftarrow \mathrm{L2norm}(W - \mathrm{Muon}(M))$ | $\mathbf{o}_t = f_W(\mathbf{q}_t)$ |

**Training Details.** We trained models at two scales using a sequence length of 32,768 tokens:

- 760M parameters: We use 24 stack blocks, with model dimension as 1536. All models are trained for 40B tokens (40,960 steps) with a sliding window of 2048 tokens and a batch size of 1 million tokens. Each experiment ran on 32 A100-40GB SXM GPUs for approximately 20 hours.

- 3B parameters: We use 25 stack blocks, with model dimension as 3072. All models are trained for 60B tokens (30,000 steps) with a sliding window of 4096 tokens and a batch size of 2 million tokens. Each experiment ran on 64 A100-40GB SXM GPUs for approximately 50-60 hours.

For both scales, we used a base learning rate of $1 \times 10^{-3}$ with a cosine decay scheduler and 1024 warmup steps. All models were randomly initialized with a standard deviation of 0.02.

**Results.** Detailed results on the RULER benchmark (Hsieh et al., 2024) are presented in Tables 7 and 8. We evaluated models on S-NIAH-1, S-NIAH-2, and S-NIAH-3 tasks, which represent varying difficulties of the single "needle in a haystack" retrieval. We also report performance on NIAH-MultiKey-1, NIAH-MultiQuery, and NIAH-MultiValue. Other RULER tasks are not reported as the full attention baseline also achieved trivial results beyond a 16K sequence length. In addition to these long context evaluations, we also report some major language model benchmark in Table 9.

Table 7: RULER benchmark results for Single Needle in a Haystack (S-NIAH) tasks. * Our method with two heads (default is four).

| | S-NIAH-1 | | | | S-NIAH-2 | | | | S-NIAH-3 | | | | Average | | | |
|---|---|---|---|---|---|---|---|---|---|---|---|---|---|---|---|---|
| Model | 4K | 8K | 16K | 32K | 4K | 8K | 16K | 32K | 4K | 8K | 16K | 32K | 4K | 8K | 16K | 32K |
| *760M parameters* | | | | | | | | | | | | | | | | |
| Transformer | 99.2 | 96.6 | 85.2 | 68.0 | 100 | 100 | 85.8 | 82.2 | 81.0 | 73.8 | 74.8 | 36.8 | 93.4 | 90.1 | 81.9 | 62.3 |
| DeltaNet + SWA | 84.0 | 85.2 | 87.8 | 86.8 | 62.8 | 29.4 | 14.2 | 7.8 | 53.8 | 21.8 | 11.2 | 5.8 | 66.9 | 45.5 | 37.7 | 33.5 |
| GLA + SWA | 51.8 | 26.2 | 14.4 | 8.6 | 55.8 | 26.4 | 15.8 | 7.8 | 58.0 | 23.8 | 16.2 | 5.0 | 55.2 | 25.5 | 15.5 | 7.1 |
| Ours | 94.8 | 53.2 | 26.0 | 14.8 | 74.0 | 28.0 | 14.2 | 7.8 | 42.8 | 26.6 | 14.4 | 6.8 | 70.5 | 35.9 | 18.2 | 9.8 |
| Ours Momentum | 95.6 | 84.8 | 83.4 | 84.8 | 91.4 | 73.4 | 22.8 | 7.8 | 82.6 | 34.8 | 16.6 | 6.6 | 89.9 | 64.3 | 40.9 | 33.1 |
| Ours Momentum* | 59.0 | 30.0 | 12.4 | 8.4 | 93.4 | 50.0 | 18.2 | 7.8 | 60.2 | 25.6 | 14.2 | 6.8 | 70.9 | 35.2 | 14.9 | 7.7 |
| Ours Muon | 98.0 | 95.0 | 92.2 | 92.4 | 86.6 | 60.2 | 17.0 | 7.8 | 49.2 | 26.2 | 10.9 | 5.2 | 77.9 | 60.5 | 40.0 | 35.1 |
| *3B parameters* | | | | | | | | | | | | | | | | |
| Transformer | 100 | 100 | 100 | 100 | 100 | 99.8 | 100 | 98.6 | 98.6 | 95.8 | 90.8 | 75.0 | 99.5 | 98.5 | 96.9 | 91.2 |
| GLA SWA | 100 | 52.8 | 26.0 | 13.2 | 100 | 51.8 | 29.6 | 14.4 | 98.0 | 54.4 | 27.6 | 12.4 | 99.3 | 53.0 | 27.7 | 13.3 |
| DeltaNet SWA | 100 | 89.6 | 76.2 | 54.8 | 100 | 76.4 | 42.2 | 17.0 | 90.6 | 57.6 | 27.4 | 13.4 | 96.9 | 74.5 | 48.6 | 28.4 |
| Ours Momentum | 99.4 | 97.0 | 98.6 | 93.4 | 100 | 75.6 | 39.6 | 15.0 | 91.8 | 63.0 | 27.8 | 13.4 | 97.1 | 78.5 | 55.3 | 40.6 |
| Ours Muon | 98.8 | 99.2 | 98.6 | 93.4 | 100 | 99.0 | 83.2 | 30.8 | 95.4 | 90.8 | 55.6 | 19.8 | 98.1 | 96.3 | 79.1 | 48.0 |

## E.3 AUTOREGRESSIVE VIDEO DIFFUSION

We fine-tune the pretrained Wan 2.1 (Wang et al., 2025a) text-to-video diffusion model into an autoregressive video diffusion model, that generates videos by iteratively denoising successive chunks of video frames.

Table 8: Performance on Multi-Key (MK-NIAH), Multi-Query (MQ-NIAH), and Multi-Value (MV-NIAH) Needle in a Haystack tasks from the RULER benchmark. * Our method with two heads (default is four).

| Model | MK-NIAH | | | | MQ-NIAH | | | | MV-NIAH | | | | Average | | | |
|---|---|---|---|---|---|---|---|---|---|---|---|---|---|---|---|---|
| | 4K | 8K | 16K | 32K | 4K | 8K | 16K | 32K | 4K | 8K | 16K | 32K | 4K | 8K | 16K | 32K |
| *760M parameters* | | | | | | | | | | | | | | | | |
| Transformer | 63.8 | 72 | 71.4 | 54 | 33.4 | 28.9 | 24 | 23.1 | 27.95 | 24 | 20.5 | 27.35 | 41.7 | 41.6 | 38.6 | 34.8 |
| DeltaNet+SWA | 41.2 | 30 | 14.6 | 8.2 | 33 | 22.45 | 7.5 | 4.3 | 32.4 | 22.8 | 9.15 | 6.6 | 35.5 | 25.1 | 10.4 | 6.4 |
| GLA + SWA | 45.4 | 28.4 | 15.8 | 6.6 | 26.1 | 17.75 | 10.2 | 5.85 | 25.4 | 16.85 | 10.1 | 6.6 | 32.3 | 21.0 | 12.0 | 6.3 |
| Ours | 60.8 | 34.6 | 16.8 | 7 | 35 | 23.65 | 14.1 | 7.45 | 20.7 | 22.05 | 12.7 | 6.85 | 38.8 | 26.8 | 14.5 | 7.1 |
| Ours Momentum | 62 | 41 | 21.2 | 10.4 | 35.3 | 24.95 | 17.7 | 8.6 | 27.9 | 23.15 | 16.65 | 8.2 | 41.7 | 29.7 | 18.5 | 9.1 |
| Ours Momentum* | 59.8 | 37.8 | 19.2 | 8.8 | 36.65 | 20.45 | 12.5 | 7.4 | 24.45 | 16.95 | 11.6 | 6.8 | 40.3 | 25.1 | 14.4 | 7.7 |
| Ours Muon | 62.8 | 46.6 | 22 | 8.6 | 37.7 | 26.55 | 15.7 | 7.1 | 28.35 | 23.15 | 13.6 | 6.85 | 42.9 | 32.1 | 17.1 | 7.5 |
| *3B parameters* | | | | | | | | | | | | | | | | |
| Transformer | 95 | 90.4 | 81.6 | 65.2 | 86.45 | 81.55 | 71.70 | 40.85 | 61.8 | 42.8 | 30.75 | 22.9 | 81.1 | 71.6 | 61.4 | 43.0 |
| GLA 3B | 78 | 45.8 | 28.6 | 14.4 | 50.05 | 28.05 | 19 | 10.7 | 29.4 | 21.4 | 16.75 | 9.9 | 52.5 | 31.8 | 21.4 | 11.7 |
| DeltaNet SWA | 75.8 | 57.4 | 34.2 | 17.8 | 66.25 | 33.05 | 21.45 | 13.45 | 43.7 | 23.2 | 18.85 | 13.2 | 61.9 | 37.9 | 24.8 | 14.8 |
| Ours Momentum | 96.2 | 59.6 | 35 | 17.2 | 87.05 | 40.25 | 25.6 | 13.2 | 88.08 | 30.65 | 21.9 | 12.3 | 90.4 | 43.5 | 27.5 | 14.2 |
| Ours Muon | 75.2 | 69.2 | 46.2 | 25.2 | 44.75 | 39.1 | 24.9 | 19 | 26.55 | 29.1 | 25.05 | 19.3 | 48.8 | 45.8 | 32.0 | 21.2 |

Table 9: Language model results across common reasoning/knowledge benchmarks (higher is better unless noted). All results are obtained through `lm-evaluation-harness` (Gao et al., 2021)

| Model | ARC-c (acc) | ARC-e (acc) | Hella. (acc_norm) | PIQA (acc_norm) | BoolQ (acc) | Wino (acc) | OpenBook (acc) | LAMBADA (acc) | SciQ (acc) | LAMBADA ppl↓ | Wiki ppl↓ | Avg. (acc) |
|---|---|---|---|---|---|---|---|---|---|---|---|---|
| **760M models** | | | | | | | | | | | | |
| Transformer | 0.247 | 0.465 | 0.405 | 0.664 | 0.588 | 0.526 | 0.190 | 0.375 | 0.864 | 22.574 | 20.750 | **0.496** |
| Transformer SWA | 0.234 | 0.479 | 0.418 | 0.663 | 0.456 | 0.502 | 0.172 | 0.407 | 0.879 | 19.537 | 21.215 | **0.496** |
| Ours Momentum | 0.247 | 0.474 | 0.413 | 0.645 | 0.611 | 0.523 | 0.188 | 0.393 | 0.861 | 20.519 | 21.300 | **0.500** |
| Ours Muon | 0.235 | 0.477 | 0.420 | 0.653 | 0.502 | 0.534 | 0.182 | 0.392 | 0.897 | 20.584 | 20.702 | **0.496** |
| GLA SWA | 0.238 | 0.475 | 0.420 | 0.649 | 0.501 | 0.528 | 0.184 | 0.393 | 0.897 | 20.621 | 20.695 | **0.502** |
| DeltaNet SWA | 0.253 | 0.466 | 0.417 | 0.650 | 0.608 | 0.515 | 0.184 | 0.386 | 0.888 | 20.415 | 23.285 | **0.502** |
| **3B+ models** | | | | | | | | | | | | |
| Transformer | 0.258 | 0.488 | 0.515 | 0.655 | 0.548 | 0.539 | 0.204 | 0.479 | 0.900 | 12.092 | 15.570 | **0.517** |
| Transformer SWA | 0.243 | 0.462 | 0.522 | 0.629 | 0.557 | 0.554 | 0.214 | 0.490 | 0.905 | 11.751 | 15.737 | **0.513** |
| Ours Momentum | 0.273 | 0.496 | 0.529 | 0.628 | 0.556 | 0.533 | 0.218 | 0.480 | 0.905 | 11.419 | 15.439 | **0.528** |
| Ours Muon | 0.263 | 0.473 | 0.523 | 0.621 | 0.522 | 0.546 | 0.210 | 0.495 | 0.894 | 10.988 | 15.313 | **0.516** |
| GLA SWA | 0.259 | 0.466 | 0.517 | 0.628 | 0.508 | 0.535 | 0.190 | 0.465 | 0.897 | 12.547 | 17.284 | **0.506** |
| DeltaNet SWA | 0.255 | 0.475 | 0.504 | 0.648 | 0.491 | 0.547 | 0.190 | 0.453 | 0.891 | 12.682 | 17.950 | **0.498** |

**Model details.** The original Wan 2.1 is a bidirectional diffusion transformer operating on the latent space of a causal video VAE, which performs 8x spatial and 4x temporal downsampling. The diffusion transformer uses a $2 \times 2 \times 1$ patchification layer to convert VAE video latents to tokens. Each block of the diffusion transformer comprises an MLP layer, a bidirectional self-attention layer for visual tokens, and a cross-attention layer for visual and text tokens.

Our primary modification is to the bidirectional self-attention. We first replace it with block-causal sliding window attention (SWA), using a window size of two chunks of video frames. We then integrate our LaCT into the same layer. We initialize learnable fast weights for LaCT. Consistent with our language modeling experiments, SWA and our test-time training mechanism are combined within each layer: Q and K vectors are rescaled and shifted before input to the test-time training operation. The outputs of SWA and the test-time training layer are summed, with a per-head learnable scalar (from a zero-initialized linear projection) applied to the latter. We do not use Muon in the fast-weight update, as it showed no significant difference in validation loss empirically. We use an fast-weight lr initialization of 0.001 by setting 'const_lr_bias' in Algorithm 1 to $\text{softplus}(\text{const\_lr\_bias}) = 0.001$. This allows small update to the fast weight in the beginning of the fine-tuning. To maintain minimal changes to the original Wan architecture, LaCT layers utilize the original RoPE from the Wan model, and we remove the SiLU activation function previously applied to queries and values.

**Datasets.** We fine-tune the model using an internal, filtered proprietary collection of videos, each accompanied by a short text prompt generated by a visual language model(Chen et al., 2024b).

**Training details.** Following (Esser et al., 2024; Wang et al., 2025a), we use time-step shifting (scale factor 3.0) and logit-normal denoising loss weighting (mean=0.5, std=1.0). We also apply an exponential moving average with a decay rate of 0.995 to the model weights. Each 5-second video

(16 FPS, 480×832 resolution) is encoded by the Wan VAE into a [21,60,104] latent representation. Denoising is performed autoregressively in chunks of three latent frames (4680 visual tokens each). We employ teacher-forcing with an interleaved noisy-clean chunk sequence (see Section 5.3).

- **1.3B Parameter Model:** For initial training on 5-second videos, noisy chunks are repeated twice. This results in sequences of 60 latent frames (14 noisy, 6 clean chunks), totaling 93,600 tokens. We finetune the model with a batch size of 64 for 5000 iterations. The base learning rate is set to $2 \times 10^{-5}$ with a linear warm-up of 1000 iterations and linear decay. Subsequently, the model is fine-tuned on 10-second video clips for 1,000 iterations. These clips correspond to 42 latent frames for the clean video portion, forming an interleaved sequence of 81 latent frames (approximately 126K tokens including noisy chunks). Training for 5-second videos takes ∼20 seconds per iteration on 64 A100 80GB SXM GPUs (or ∼10 seconds on 64 H100 80GB SXM GPUs).

- **14B Parameter Model:.** To manage GPU memory usage, noisy chunks are not repeated in this setting. We train the model on five-second videos with a batch size of 64 for 5000 iterations with a base learning rate of $5 \times 10^{-6}$, and use a sequence parallel size of 2 GPUs. This phase takes ∼80 seconds per iteration on 64 A100 GPUs. The model is then fine-tuned on 8.8-second video clips (36 latent frames for the clean portion) for an additional 600 iterations, using sequence parallelism (4 GPUs). This fine-tuning takes ∼80 seconds per iteration on 64 H100 GPUs.

**Baselines.** We compare our method with three baselines: sliding window attention, Mamba2 (Dao & Gu, 2024) with sliding window attention, and full block-wise causal attention, where the window attention in the baselines is implemented the same as in our model. For the Mamba2 layer, we follow (Wang et al., 2024) to apply the original projected $k$, $q$, and $v$ as $B$, $C$, and $x$, respectively. The Mamba2's state is updated token-by-token, we revert the state after processing a noise chunk of frames to ensure only clean chunk state updates propagate. The full block-wise causal attention baseline is implemented with FlexAttention (He et al., 2024).

**Evaluation.** We compute validation loss for all models on a collection of 2,000 videos after 5,000 training iterations by computing the denoising loss at five timesteps (550, 650, 750, 850, 950). The denoising losses are measured with respect to each video frame chunk and plotted in Figure 6. Figure 6(a) compares validation loss (up to 5s videos) of LaCT against SWA, Mamba2 with SWA, and full block-wise causal attention. Our LaCT is comparable to full attention and outperforms other baselines. Figure 6(b) shows comparisons with the SWA baseline using different window sizes for both our method and the baseline (up to 5s videos). The default window covers six latent frames (two chunks). An additional experiment used a four-frame window. Results indicate that increasing window size from four to six frames improves validation loss, but this improvement is smaller than that achieved by incorporating LaCT. Figure 6(c) presents validation loss (up to 10s videos) after fine-tuning LaCT and the SWA baseline on 10-second videos for 1,000 iterations.

Generated video samples from our model are provided in an appended folder. Each video chunk is sampled following the original Wan method, using a UniPC (Zhao et al., 2023) sampler with 50 steps, classifier-free guidance of 5.0, and a timestep shift of 3.0.

**Results on VBench.** VBench (Huang et al., 2024) offers a comprehensive suite of metrics for video generation. Using 942 prompts from the standard set, we generate two videos per prompt for our 1.3B model and one video per prompt for our 14B model. Summary scores are reported in Table 10, with detailed results across 16 dimensions in Table 12.

Because the standard prompts are short, the Wan team also provides augmented prompts with richer descriptions. We evaluate on this augmented set as well, reporting summary results in Table 11 and detailed scores in Table 13.

Overall, we observe two key trends: (1) autoregressive models do not yet match their full-sequence diffusion counterparts, and (2) among autoregressive approaches, our test-time training method consistently outperforms sliding-window and full-attention variants in temporal quality, semantic score, and total score, while maintaining parity on other metrics. The performance gap is even larger under the standard prompt set.

Table 10: Summarized VBench scores using the standard prompt set. (higher is better).

| Type | Method | Temporal Quality | Frame Quality | Text Alignment | Quality Score | Semantic Score | Total Score |
|------|--------|------------------|---------------|----------------|---------------|----------------|-------------|
| Full Seq | Original Wan (1.3B) | 91.94% | 64.27% | 25.06% | 83.59% | 67.13% | 80.30% |
| AR | Ours (1.3B) | 92.50% | 63.60% | 24.75% | 82.51% | 62.17% | 78.44% |
| AR | Transformer SWA (1.3B) | 91.14% | 63.03% | 24.77% | 81.62% | 60.09% | 77.31% |
| AR | Transformer (1.3B) | 92.32% | 62.19% | 24.78% | 82.28% | 60.40% | 77.90% |
| Full Seq | Original Wan (14B) | 93.08% | 64.52% | 25.79% | 84.31% | 69.53% | 81.35% |
| AR | Ours (14B) | 92.79% | 63.30% | 25.67% | 82.89% | 65.86% | 79.49% |

Table 11: Summarized VBench scores using the WAN-augmented prompt set. (higher is better).

| Type | Method | Temporal Quality | Frame Quality | Text Alignment | Quality Score | Semantic Score | Total Score |
|------|--------|------------------|---------------|----------------|---------------|----------------|-------------|
| Full Seq | Original Wan (1.3B) | 92.20% | 66.52% | 26.16% | 84.48% | 75.98% | 82.78% |
| AR | Ours (1.3B) | 93.97% | 66.58% | 26.04% | 84.41% | 74.25% | 82.38% |
| AR | Transformer SWA (1.3B) | 93.20% | 66.53% | 26.09% | 84.48% | 73.00% | 82.18% |
| AR | Transformer (1.3B) | 93.62% | 66.37% | 26.00% | 84.57% | 72.95% | 82.25% |
| Full Seq | Original Wan (14B) | 93.18% | 66.85% | 26.27% | 85.13% | 77.07% | 83.52% |
| AR | Ours (14B) | 94.60% | 66.17% | 26.16% | 84.99% | 72.85% | 82.57% |

### E.4 EXPERIMENT DETAILS IN FIGURE 1

Fig. 1(c) shows results for training a 760M-parameter LaCT language model. We employ a SwiGLU MLP fast weight with the Muon test-time optimizer. To scale the fast weight size, we fix the intermediate dimension of the fast-weight MLP to match the head dimension, then increase the head dimension from 128 to 1536 while proportionally decreasing the number of heads to maintain a constant total model dimension. Validation loss is computed on the last 2,048 tokens of each 32,768-token sequence, averaged over 76K sequences from the Book3 dataset.

Fig. 1(d) uses the object-level novel view synthesis experiment. All models consist of 14 stacked blocks with a fixed model dimension of 768 and were trained for 167 billion tokens. Training time (wall-clock) is measured on an A100-40GB SXM GPU.

## F DETAILS FOR MAMBA BASELINES

Mamba is an efficient model architecture, it is logically similar to a linear TTT taking per-token linear update rule of the fast weight (i.e., state in Mamba's context). Thus it serves as a baseline to understand the gap between the chunk-wise update and per-wise update in Fig. 8 and Fig. 6. In this section, we detailed the experimental setup.

We take the official Mamba-2 implementation[3] in all our experiment. The original Mamba-2 has multiple components, and we largely simplify its implementation to keep a measurable architecture while still maintaining the performance. In detail, our Mamba-2's formulation in experiment is:

$$X, B, C, \delta = \text{Linear}(u)$$
$$\delta = \text{softplus}(\delta + \delta_{init})$$
$$H_t = \exp(-\delta_t)H_{t-1} + \delta_t B_t^T X_t$$
$$y_t = C_t H_t \tag{22}$$

where $X$, $B$, $C$ is of shape $(L, d)$, and $\delta$ is of shape $(L, 1)$. $H_t$ is a matrix state of shape $(d, d)$.

---

[3]https://github.com/state-spaces/mamba

Table 12: VBench subscores using standard text prompt set. (higher is better).

| Models | Type | Subject Consistency | Background Consistency | Temporal Flickering | Motion Smoothness | Dynamic Degree | Aesthetic Quality | Imaging Quality | Object Class |
|---|---|---|---|---|---|---|---|---|---|
| Full Seq | Original Wan (1.3B) | 95.36% | 96.60% | 99.42% | 98.24% | 63.19% | 61.10% | 67.44% | 76.98% |
| AR | Ours (1.3B) | 93.94% | 94.47% | 96.98% | 98.20% | 74.31% | 59.81% | 67.40% | 69.66% |
| AR | Transformer SWA (1.3B) | 92.23% | 91.49% | 97.62% | 98.11% | 74.31% | 58.59% | 67.46% | 66.10% |
| AR | Transformer (1.3B) | 92.90% | 93.68% | 98.05% | 98.00% | 77.08% | 58.36% | 66.02% | 67.76% |
| Full Seq | Original Wan (14B) | 95.04% | 96.89% | 99.28% | 98.44% | 70.83% | 62.27% | 66.78% | 81.96% |
| AR | Ours (14B) | 93.57% | 94.69% | 97.65% | 98.31% | 76.39% | 59.83% | 66.78% | 73.26% |

| Type | Models | Multiple Objects | Human Action | Color | Spatial Relationship | Scene | Appearance Style | Temporal Style | Overall Consistency |
|---|---|---|---|---|---|---|---|---|---|
| Full Seq | Original Wan (1.3B) | 60.86% | 77.50% | 91.91% | 72.69% | 19.44% | 20.24% | 23.62% | 23.64% |
| AR | Ours (1.3B) | 44.74% | 73.50% | 81.39% | 66.01% | 21.69% | 20.41% | 23.12% | 22.92% |
| AR | Transformer SWA (1.3B) | 35.75% | 75.50% | 81.49% | 58.91% | 21.18% | 20.26% | 23.14% | 22.86% |
| AR | Transformer (1.3B) | 37.08% | 70.00% | 85.17% | 61.99% | 19.73% | 20.22% | 23.06% | 23.12% |
| Full Seq | Original Wan (14B) | 62.80% | 77.00% | 90.24% | 72.96% | 27.33% | 21.36% | 23.38% | 24.99% |
| AR | Ours (14B) | 49.85% | 85.00% | 86.18% | 63.62% | 22.24% | 21.86% | 23.28% | 24.51% |

Table 13: VBench subscores using the WAN-augmented prompt set. (higher is better).

| Type | Method | Subject Consistency | Background Consistency | Temporal Flickering | Motion Smoothness | Dynamic Degree | Aesthetic Quality | Imaging Quality | Object Class |
|---|---|---|---|---|---|---|---|---|---|
| Full Seq | Original Wan (1.3B) | 94.87% | 96.64% | 99.31% | 98.55% | 65.28% | 65.71% | 67.34% | 87.22% |
| AR | Ours (1.3B) | 92.87% | 94.77% | 97.49% | 98.19% | 86.11% | 65.36% | 67.81% | 88.13% |
| AR | Transformer SWA (1.3B) | 91.69% | 94.63% | 98.75% | 98.22% | 83.33% | 64.81% | 68.26% | 88.37% |
| AR | Transformer (1.3B) | 92.26% | 94.68% | 98.39% | 98.35% | 84.72% | 64.72% | 68.02% | 82.04% |
| Full Seq | Original Wan (14B) | 95.05% | 97.13% | 99.27% | 98.55% | 70.83% | 66.18% | 67.52% | 91.38% |
| AR | Ours (14B) | 92.39% | 95.35% | 98.33% | 98.21% | 90.28% | 64.18% | 68.17% | 85.05% |

| Type | Method | Multiple Objects | Human Action | Color | Spatial Relationship | Scene | Appearance Style | Temporal Style | Overall Consistency |
|---|---|---|---|---|---|---|---|---|---|
| Full seq | Original Wan (1.3B) | 73.21% | 95.00% | 90.59% | 74.53% | 45.09% | 21.58% | 23.02% | 25.36% |
| AR | Ours (1.3B) | 63.03% | 92.00% | 89.83% | 70.95% | 47.09% | 21.35% | 22.70% | 25.45% |
| AR | Transformer SWA (1.3B) | 62.65% | 95.00% | 86.24% | 66.75% | 41.79% | 21.31% | 22.72% | 25.56% |
| AR | Transformer (1.3B) | 66.23% | 96.00% | 82.58% | 72.45% | 40.70% | 21.07% | 22.93% | 25.87% |
| Full Seq | Original Wan (14B) | 72.87% | 94.00% | 90.45% | 75.29% | 49.85% | 22.66% | 21.83% | 25.37% |
| AR | Ours (14B) | 64.33% | 93.00% | 83.47% | 68.03% | 43.24% | 22.34% | 21.94% | 25.75% |

Transferring the above formula to a standard linear-attention / TTT / DeltaNet notations, it is equivalent to:

$$
\begin{aligned}
V, K, Q, lr &= \text{Linear}(input) \\
lr &= \text{softplus}(lr + lr_{init}) \\
W_t &= \exp(-lr_t)W_{t-1} + lr_t K_t^T V_t \\
O_t &= Q_t W_t
\end{aligned}
\tag{23}
$$

We will denote the above equations as $O = \text{Mamba}(input)$.

We use the multi-head design as in Transformer's multi-head attention. Multiple independent Mamba-2 layer are run in parallel and their outputs are concatenated. Suppose the number of heads is $nh$, the formula is:

$$
\begin{aligned}
O^k &= \text{Mamba}^k(input) \\
O &= [O^1, \ldots, O^{nh}]
\end{aligned}
\tag{24}
$$

where each $\text{Mamba}^k$ is a Mamba with its own parameters. In Mamba-2's terminology, this design is equivalent to setting the number of 'groups' to be the same as the number of heads.

For the novel view synthesis task, we take a bidirectional Mamba over the input image tokens. In detail, we take two independent multi-head Mamba with one reading from left to right and the other reading from right to left. The bidirectional model builds a better connection among input tokens and also doubles the state size. We use a similar 'apply' operation as in LaCT that only updates

the state for input tokens, and the state is static for the target tokens. We also tested with 'update' for the target image tokens, but it empirically leads to worse results. We use a head dimension of 192 and 8 heads. The overall state size, 8 (num heads) $\times$ $192^2$ (head dim) $\times$ 2 (bidir), matches LaCT with a standard large-chunk large-weight linear attention of dimension 768 (768 input dim $\times$ 768 intermediate dim in Fig. 8. We take $lr_{init} = -4.6$, which corresponds to a 0.01 initialized learning rate (i.e., $\text{softplus}(-4.6) = 0.01$).

For the autoregressive video diffusion task, we apply a unidirectional Mamba over the flattened video tokens. As mentioned in Sec 6.3, we follow (Wang et al., 2024) to inherit the Wan's self-attention projected $k$, $q$, and $v$ as $B$, $C$, and $x$ in the Mamba layer, respectively. Unlike in the NVS task, each token will 'update' the state, which will be 'applied' to the current output and future tokens. Our Mamba uses 12 heads, each of dimension 128, matching the original multi-head self-attention in Wan. The overall state size is 12 (num heads) $\times$ $128^2$ (head dim). We take $lr_{init} = -4.6$, which corresponds to a 0.01 initialized learning rate.

## G    DETAILS OF LaCT CONTEXT PARALLELISM IMPLEMENTATION

Context Parallelism(CP) partitions the input sequence along its sequence length dimension and distributed the shards across multiple devices for parallel computing. The feed-forward layer and window attentions are local operations thus support CP naively. Our large-chunk Test-Time Training (TTT) approach facilitates CP by sharding tokens within each large chunk.

Within our large-chunk TTT mechanism, the per-token *apply* operation naively supports CP due to its independent nature. The *update* allows CP by shading tokens within a chunk over multiple devices. This CP can be easily implemented by adding a few lines of distributed all-reduce-sum after computing the local fast weight gradients on each device, logically the same as the Distributed Data Parallellism. Note that the distributed all-reduce-sum is a differentiable operator and its backward is all-reduce-sum over the gradient, thus the network can be trained end-to-end. Algorithm 3 presents the pseudocode detailing this intra-chunk context parallelism specifically for the large-chunk TTT *update* operation. We employed this parallelism in our view synthesis experiments, handling maximum chunk sizes exceeding half a million tokens and maximum sequence lengths over one million tokens during training.

## H    DETAILS OF LaCT TENSOR PARALLELISM IMPLEMENTATION

Beyond Context Parallelism, our large-chunk Test-Time Training (TTT) mechanism also supports Tensor Parallelism (TP). This is primarily achieved by sharding the TTT heads across multiple devices, a strategy similar to that employed in methods like DeepSpeed Ulysses (Jacobs et al., 2023).

Specifically, while static feed-forward layers in the model might process inputs sharded along the sequence dimension (Context Parallelism), for the TTT operations within our LaCT layer, the data undergoes a gather-then-scatter transformation. Input tensors (Q, K, V, and learning rates for TTT) that are initially sharded by sequence length are first gathered along the sequence dimension to reconstruct the full sequence context on each device within the tensor-parallel group. Then, these full-sequence tensors are scattered along the head dimension. As a result, each device processes the complete sequence but operates on only its assigned subset of TTT heads during the TTT *update* and *apply* iterations. The reverse transformation (gather heads, scatter sequence) is applied to the output of TTT operation. Algorithm 4 provides pseudocode detailing this tensor parallelism implementation, omitting minor details like padding. While this gather-then-scatter method effectively enables head-sharded tensor parallelism, more sophisticated communication strategies (Fang & Zhao, 2024; Jacobs et al., 2023) could potentially be employed to further optimize communication overhead.

We utilized this tensor parallelism strategy in our autoregressive video generation experiments, sharding, for example, four TTT heads across four local GPUs. This enabled us to train 14-billion-parameter diffusion models with sequence lengths exceeding 100K tokens.

**Algorithm 3** Large Chunk Test-Time Training Layer with Context Parallel Sharded inside chunk Pseudocode

```python
def update(fast_weight, k, v, lr, cp_group, use_muon=True):
    """
    Fast-weight update for a SwiGLU MLP using a context-parallel chunk.

    Args:
        fast_weight : tuple(w1, w2, w3) with shapes: w1, w3: [b, d, dh]; w2: [b, dh, d]
        k, v : key / value tensor of shape [b, l, d]
        lr: : per-token learaning rates of shape [b, l, 3] -> (lr1, lr2, lr3)
        cp_group : process group metadata for context parallelism
        use_muon : weather to apply Muon to orthogonalize the update

    Note:
        The input tensors k, v, lr are assumed to be already partitioned (sharded)
            along the sequence
        dimension over multiple devices. l represents the local sharded sequence length
            on each device.
        The total effective chunk size processed is l * cp_group.size.
    """

    # Forward with k:
    gate_before_act = matmul(k, w1) # [b, l, dh] = [b, l, d] x [b, d, dh]
    hidden_before_gate = matmul(k, w3) # [b, l, dh] = [b, l, d] x [b, d, dh]
    hidden = silu(gate_before_act) * hidden_before_gate

    # Backward:
    dhidden = matmul(v, w2.transpose(-1, -2)) # [b, l, dh] = [b, l, d] x [b, d, dh]
    dhidden_before_gate = dhidden * silu(gate_before_act)
    dgate = dhidden * hidden_before_gate
    dgate_before_act = silu_backprop(dgate, gate_before_act)

    # Compute gradients:
    w2.grad = -matmul(hidden.transpose(-1, -2), v * lr2) # [b, dh, d] = [b, dh, l] x [
        b, l, d]
    # [b, d, dh] = [b, d, l] x [b, l, dh]
    w1.grad = -matmul((k * lr1).transpose(-1, -2), dgate_before_act)
    w3.grad = -matmul((k * lr3).transpose(-1, -2), dhidden_before_gate)

    # [Standard forward pass and local backward gradient computations are performed
        above,
    # resulting in local w.grad for each device.]

    ################################################################################
    # BEGIN CONTEXT PARALLELISM SPECIFIC MODIFICATION: Global Gradient Aggregation
    # The following AllReduce operation is the key step introduced for context
    # parallelism. Operations before this point compute local gradients; operations
    # after this point use the globally aggregated gradients.
    ################################################################################
    for w in fast_weight:
        w.grad = distributed_all_reduce(w.grad, cp_group, op="SUM")
    ################################################################################
    # END CONTEXT PARALLELISM SPECIFIC MODIFICATION.
    # Subsequent operations (Muon, weight updates) now use the globally summed w.grad.
    # The formulas for these subsequent operations remain the same as in a
    # non-parallel version, but they act upon these aggregated gradients.
    ################################################################################

    # Weight update
    if use_muon:
        for w in fast_weight:
            w.grad = zeropower_via_newtonschulz5(w.grad)
    for w in fast_weight:
        w = (w - w.grad) / (w - w.grad).norm(dim=1) * w.norm(dim=1)

    return fast_weight
```

## I  LIMITATION

One limitation of our method is the absence of rotation invariance. Unlike softmax attention and linear attention, which remain invariant under uniform rotations of queries and keys (a property leveraged by relative positional encodings such as RoPE Su et al. (2023)), our SwiGLU and Linear Fast Weight components do not exhibit this property. The practical implications of this absence remain underexplored.

On the language modeling task, some key aspects are not explored due to computation limitation. These aspects include the reasoning capacity of our LaCT model and also the scalability regarding

**Algorithm 4** Large Chunk Test-Time Training Layer with Tensor Parallelism by sharding heads Pseudocode

```python
def gather_scatter(x, gather_dim, scatter_dim, process_group=cp_group):
    """
    Gathers tensor x along gather_dim across process_group,
    then scatters the result along scatter_dim to each device locally.
    Example: Transform [B, N_full, L_local, D] with gather_dim=2, scatter_dim=1
             to [B, N_local, L_full, D] on each device.
    """
    x = all_gather(x, gather_dim, process_group)

    # Calculate slicing indices for the scatter operation
    local_rank, group_size = process_group.rank, process_group.size
    scatter_stride = x.size(scatter_dim) // group_size
    start_idx = local_rank * scatter_stride
    end_idx = (local_rank + 1) * scatter_stride

    # Slice the tensor to get the local shard for the current device
    x = slice_tensor(x, scatter_dim, start_idx, end_idx)
    return x

######### MultiHead LaCT Layer with Tensor Parallelism (sharding TTT heads) #########
# Input:
# x: input sequence sharded by sequeunce length (CP). Shape [b, l, d], b is the batch
#     dim, l is local sequence length, d is model dimension.
# fast_weight: tuple of sharded initial fast weights, sharede among heads. (w1, w2,
#     w3); w1, w3 of shape [nh, d, dh], w2 of shape [nh, dh, d]. nh: number of local
#     heads.

qkv = silu(LinearQKV(x)) # [b, l, d * 3]
qkv = rearrange(qkv, `b l (nh hd) -> b nh l hd`, nh=num_heads).split(3, dim=-1)
q, k = q / q.norm(-1), k / k.norm(-1)
lr = softplus(LinearLR(x) + const_lr_bias) # [b, l, 3 * num_heads]
lr = rearrange(lr, `b l (nh 3) -> b nh l 3`, nh=num_heads)

################################################################################
# BEGIN TENSOR PARALLELISM SPECIFIC TRANSFORMATION
# Gather along Sequence Length (dim 2), then Scatter along Head Dimension (dim 1).
# Transforms [b, nh_full, l_local, X] -> [b, nh_local, l_full, X]
# Each device now has the full sequence for a subset of heads.
q, k, v, lr = map(lambda x: gather_scatter(x, gather_dim=2, scatter_dim=1), (q, k, v,
    lr))
# END TENSOR PARALLELISM SPECIFIC TRANSFORMATION
################################################################################

# [b, nh_local, l_full, X]
o_local_heads = ... # Placeholder for actual TTT computation on sharded heads
o_local_heads = RMSNorm(o_local_heads) # per-head norm

################################################################################
# BEGIN TENSOR PARALLELISM SPECIFIC REVERSE TRANSFORMATION
# Gather along Head Dimension (dim 1), then Scatter along Sequence Dimension (dim 2).
# Transforms [b, nh_local, l_full, X] -> [b, nh_full, l_local, X]
# This reconstructs the full head dimension but shards sequence back.
o = gather_scatter(o_local_heads, gather_dim=1, scatter_dim=2)
# END TENSOR PARallelism SPECIFIC REVERSE TRANSFORMATION
################################################################################
o = rearrange(o, `b nh l hd -> b l (nh hd)`, nh=num_heads)
o = LinearOutput(o)

return o
```

the parameter size. Previous papers showed that a main weakness of the state-based model (where LaCT belongs to) is its reasoning ability. However, the reasoning ability is only gained with certain amount of training compute thus it is beyond our budget.

Lastly, for the autoregressive video diffusion, it is hard to find a reliable and distinguishable metric to measure the model's scalability. It is in contrast to the language modeling with perplexity (i.e., log likelihood loss) and the novel-view synthesis with PSNR. We show the validation loss in our paper and it is a common choice in evaluating the scalability of video generation. This is a general problem for the video generation evaluation and is not specific to our paper.

## J   LLM USAGE

We employ large language models (LLMs) in three main scenarios. During the experimental stage, we use LLMs to generate portions of tedious code based on clear instructions and guidelines to acclerate experimental iteration. During the writing stage, we leverage LLMs to polish the text and identify grammatical issues. Finally, we also use LLMs to generate Python code for plotting figures with `matplotlib`.

