# OpenReview forum: "Test-Time Training Done Right"
_ICLR.cc/2026/Conference — ICLR 2026 Poster_

### Official Review · Reviewer_21cR · 2025-10-31

**Soundness:** 3
**Presentation:** 3
**Contribution:** 2
**Rating:** 4
**Confidence:** 4

**Summary:**

The paper addresses the problem of low FLOP utilization with TTT layers. They proposes Large-Chunk Test-Time Training (LaCT): updating fast weights over very large chunks compared to prior work to reach high FLOP utilization. As updates of fast weights treats token in the current chunk as sets, TTT layers are paired with sliding window attention to recover locality. The method is validated on novel view synthesis (NVS, up to ~1M tokens), language modeling (per-position loss & RULER/S-NIAH retrieval), and autoregressive video diffusion showing competitive performance with linear time models and Transformers.

**Strengths:**

1. The suggested solution of combining large chunks with SWA is effective, simple and easy to implement.
2. Evaluation is very robust spanning multiple modalities and tasks, comparing to relevant baselines.

**Weaknesses:**

1. Combining linear time sequence models with attention is not new and has been shown to improve the performance, see e.g., [1], although not motivated from low throughput.
2. The method strongly relies on large chunk sizes yet the effect of chunk sizes on performance (while maintaining high throughput) is not clear from the results.

[1] An Empirical Study of Mamba-based Language Models

**Questions:**

1. Is it possible to ablate the effect of the chunk size on performance?
2. Can you clarify the end-to-end complexity where the chunk size equals the full sequence length?

---

> ### Author Response · Authors · 2025-11-22
> **First round rebuttal to reviewer 21cR; Part 1**
>
> We appreciate the reviewer's efforts and constructive feedback, and here we address your question.
>
>
> ## "Combining linear time sequence models with attention is not new and has been shown to improve the performance, see e.g., [1], although not motivated from low throughput."
>
> We agree that there are many linear time sequence models, and many hybrid models that combine modern RNN with softmax attention.  And we have disccused with them in the paper, e.g. the first two paragraph in the related work sections.
>
> We mainlly address the low FLOPS utilization problem seen in TTT layers, which you also mentioned in your summarization.
> We address so with a simple strategy:  using large chunk recurrence, rather than staying with per-token or fine grained recurrence adopted by previous work. And we show that such large chunk recurrence has multiple advantages not only in hardware efficiency through higher compute intensity and more parallelsim, but also in better supports for non-causal datas, e.g. images,  easier implementations (pytorch only implementation suffice without relying on customized kernels), and also easier integration with more compliacted state update equations, e.g. muon optimizer.
>
> \[1\] An Empirical Study of Mamba-based Language Models
>
> ## Effects of chunk-size
>
> I agree with you. Here we provide analysis of effects of chunk size.
>
>
> We measured the hardware utilization of the large-chunk TTT update used in our language modeling experiments across different chunk sizes, while keeping the following fixed: batch size = 4; sequence length = 32,768; SwiGLU intermediate multiplier = 2.
>
> The table below reports the achieved TFLOP/s on H100 SXM5 as we vary chunk size and head dimension:
>
> | head_dim ↓ / chunk_size → | **1024** (TFLOP/s) | **2048** (TFLOP/s) | **4096** (TFLOP/s) | **8192** (TFLOP/s) | **16384** (TFLOP/s) |
> |---------------------------|---------------------|---------------------|---------------------|---------------------|----------------------|
> | **512**                   | 87.1                | 147.8               | 283.2               | 321.8               | 347.7                |
> | **1024**                  | 288.9               | 400.9               | 455.6               | 490.2               | 484.6                |
>
>
> For performance: we are running experiments with different chunk sizes and will update you once doen.
>
>
>
> ## Can you clarify the end-to-end complexity where the chunk size equals the full sequence length?
>
> When the chunk size equals the full sequence length, LaCT still has linear computational complexity with respect to the number of tokens.
>
> In this case, we use the key–value pairs from all input tokens to update the fast weight once, which has linear computational cost. After this update, we apply the updated fast weight to all query tokens, which also has linear computational cost.
>
> **FLOPs breakdown:**
> Suppose the head dimension is $D$, the hidden dimension of the MLP fast weight is $H$ (which is typically $D$ or $2D$), and the sequence length is $N$.
>
> Computing the fast-weight gradient involves 6 matmuls, for a total cost of:$12 * D * H * N$ FLOPs
>
> Applying the updated fast weight to all query tokens involves 3 matmuls, for a cost of $6 * D *H * N$ FLOPs.
>
> So the total FLOPs is $18 *  D * H * N$
> We presented a variant of this derivation in Equation (14) of the paper.
>
>
> Please let me know if any further clarification would be helpful.

---

> > ### Comment · Reviewer_21cR · 2025-11-26
> >
> > I am possibly misunderstanding but can you clarify how is the complexity linear when the chunk size is set as the full sequence length? doesn't window attention with a single chunk size entails quadratic complexity?
> >
> > Are the window size used in attention layers and chunk size used for updates not the same? e.g. for language modeling experiment the main text implies they are. For hybrid models as in [1] there is no tying between the attention window or chunk size which maintains linear complexity.
> >
> > [1] An Empirical Study of Mamba-based Language Models

---

> > > ### Author Response · Authors · 2025-11-27
> > > **Reply for reviewer 21cR**
> > >
> > > Thank you for the engagement, happy to clarify.
> > >
> > > ## Regarding Chunk-size:
> > >
> > >
> > > Yes, you mis-understood.
> > > The test-time training chunk size can be **different** from the attention window size used in the model architecture. For example:
> > >
> > >
> > > ### For the novel view synthesis experiment,
> > > * Chunk size: all tokens from all input images
> > >
> > > * Window attention size: number of tokens within each image (attention is applied per image independently)
> > >
> > >
> > > For example, in Fig. 4(c) with 128 input images:
> > >
> > > * chunk size:  $128 \times 8040=1,034,240$
> > > * window attention size;  $8040$
> > >
> > > Chunk size is much larger, but compute remains linear because attention never exceeds the per-image window.  And the window attention here is to help modelling locality within each image.
> > >
> > >
> > > ### For the language model experiment
> > >
> > > We usually set chunk size equals to window attention size. However, **they do not need to match**. The only constraint is that: chunk-size <= window-attention size
> > >
> > > Otherwise, “bubbles” appear, there will be regions of the sequence that cannot access earlier context. For example, in the language model case, if sliding window size = 2048 and LaCT chunk size = 4096.  Then, tokens at positions 3072–4096 cannot access tokens at positions 0–1024.
> > >
> > > Thus, for causal 1D sequences, full-sequence chunking is not meaningful unless the window is also full-sequence (i.e., equivalent to full attention).
> > >
> > > In our ablation on chunk in previous reply, we fixed window size = 2048 and varied chunk size ∈ {512, 1024, 2048}.
> > > All satisfy chunk ≤ window, so there are no bubbles and FLOPs remain nearly identical.
> > >
> > >
> > >
> > > ## Relation to [1]
> > >
> > > The cited work [1] did a great large-scale study for both pure Mamba (linear-time) and hybrid models combining Mamba with full attention.
> > > In all our experiments, the underlying attention remains local-window–based, ensuring that LaCT always maintains linear compute complexity, regardless of chunk size.
> > >
> > > Please let me know if any part would benefit from further clarification.
> > >
> > > [1] An Empirical Study of Mamba-based Language Models

---

> ### Author Response · Authors · 2025-11-23
> **First round rebuttal to reviewer 21cR; Part 2**
>
> ## Effects of chunk size for performance
>
> Thanks for raising this insightful question. You and reviewer vmAu all asked this question, which I appreciate a lot.
>
>  We conducted an ablation study on the LaCT chunk size in both the language modeling and novel-view-synthesis (NVS) experiments. In all experiments, we fix the window-attention size and vary only the LaCT chunk size.  These two experiments shows that the chunk size has minimal impact on performance.
>
> **Ablation: Chunk size in language modeling**
> We repeat the LM setup from the paper with a smaller compute budget: sequence length 16,384 and 20B training tokens (vs. 32k sequence length trained for 40B in Fig. 7–8). We evaluate per-position loss over 2.5B tokens, as done in the main paper. The per-position loss (length 16,384) is divided into 8 equal segments, and we report the average loss per segment.
>
> Results below show negligible differences across chunk sizes:
>
>
> | Val loss at Segment | Chunk Size 2048 | Chunk Size 1024 | Chunk Size 512 |
> | --- | --- | --- | --- |
> | 1-2,048 | 2.778 | 2.777 | 2.776 |
> | 2,049-4,096 | 2.690 | 2.688 | 2.689 |
> | 4,097-6,144 | 2.682 | 2.680 | 2.682 |
> | 6,145-8,192 | 2.678 | 2.676 | 2.679 |
> | 8,193-10,240 | 2.676 | 2.674 | 2.677 |
> | 10,241-12,288 | 2.675 | 2.672 | 2.676 |
> | 12,289-14,336 | 2.674 | 2.671 | 2.675 |
> | 14,337-16,384 | 2.673 | 2.671 | 2.674 |
>
>
>
> **Ablation: Chunk size in Novel-View-Synthesis experiment**
> We also test chunk size sensitivity on the object-level NVS experiment using a smaller model (8 blocks instead of 14 as in Figure 7, 8 of the paper). Similarly we. train the model with 8 input and 8 output views.  Each image has resolution of $256\times256$, resulting 1024 tokens per image, 16384 tokens per sequence.  We train each model for 167B tokens, and report the novel view PSNR on evaluation set with 8 input images.  Again, performance is largely unchanged across chunk sizes:
>
>
> | Metric \ Chunk-size | 8 images | 4 images | 2 images | 1 images |
> |--------|---------|---------|---------|---------|
> | PSNR   | 28.50   | 28.40   | 28.48   | 28.41   |

---

### Official Review · Reviewer_YxH9 · 2025-10-31

**Soundness:** 3
**Presentation:** 2
**Contribution:** 3
**Rating:** 6
**Confidence:** 3

**Summary:**

This paper proposes Large-Chunk Test-Time Training (LaCT), a novel test-time training (TTT) method that updates “fast weights” in large chunks rather than small batches. TTT dramatically improves GPU utilization and scalability of nonlinear state size without requiring custom CUDA kernels.

LaCT integrates windowed attention for local structure, TTT for long-range context, and is flexibly compatible with N-dimensional data. The method is validated on novel view synthesis, language modeling, and autoregressive video diffusion.

**Strengths:**

- **Extremely thorough experimentation**: LaCT is tested on three different downstream tasks across two different modalities (text, image/video).  Throughput, scaling, and efficiency experiments are also provided to exhibit the feasibility of LaCT.
- **Clear writing and strong figures**: Lots of preliminaries and related works are provided, as well as a clear linear flow of the paper introducing the research problem and solution (poor TTT GPU utilization caused by small-chunk updated).
- **Scalability and practicality**: Runs efficiently in PyTorch with strong FLOP utilization, making it generally adaptable - especially given the lack of  reliance on custom CUDA kernels.

**Weaknesses:**

- **Core results deferred to appendix**: Most quantitative comparisons and ablations are in the appendix, leaving the main paper heavy on discussion. For example, the experiments section provides very surface-level details regarding datasets, training details, etc. which are pushed down to the appendix. This makes the actual quantitative impact of LaCT on downstream datasets less apparent. For example, Tables 3 and 4 in the appendix highlight the near identical performance between full attention and LaCT, however without any discussion highlighting that LaCT is more efficient, the takeaway is unclear.
- **Sparse dataset/training details for autoregressive video diffusion**: The authors very briefly mention using a an internal, proprietary video dataset with minimal transparency, limiting reproducibility.
- **Limited comparison to alternative TTT methods**: Comparisons against other TTT methods (e.g., Titans, Atlas, which are cited in the main paper under Behrouz) are not provided. While they may be much too inefficient to compare to the scale and downstream tasks of LaCT, explicit quantitative results highlighting this point would be useful.

**Questions:**

1. How does LaCT compare against previous TTT methods, given its much higher propensity for scalability?
2. Why was LaCT trained on proprietary data when any video diffusion dataset could have sufficed?

Please address question in the Weaknesses section as well. Overall while the paper is quite dense, the method is interesting, applicable, and thoroughly ablated and tested.

---

> ### Author Response · Authors · 2025-11-22
> **First round rebuttal to reviewer YxH9; Part 1**
>
> We thank the reviewer for their appreciation and constructive feedbacks, here is our first round rebuttal.
>
> ## Core results deferred to appendix.
> Thank you for pointing this out. We agree that some dataset and training details can be moved to the appendix and that several key ablations should be brought into the main paper.
>
> Regarding your specific comment on Tables 3 and 4: this is an excellent observation. For readers familiar with novel-view synthesis, the near-identical performance to full attention is a strong result, but without highlighting the efficiency gains, the takeaway can be unclear.  We will explicitly emphasize the efficiency advantage of LaCT in these tables.  Additionally, since Table-3 and Table-4, and Table-5 basically gives a detailed results for every experiment shown in Figure-4 of the main paper.  I can also add cross-reference in Figure-4 and table-3,4,5.
>
> ## Sparse dataset/training details for autoregressive video diffusion.
>
> We appreciate the concern. We do provide detailed training information in Appendix E.3, and we have included all training code in folder `lact_ar_video/` of the attached supplementary material.
>
> While we cannot release the proprietary dataset, we can offer additional transparency and details, and will include them in the paper.
>
> The dataset consists of a large-scale collection of professionally produced video clips, across diverse themes such as lifestyle, urban scenes, nature, business activities, and cinematic b-roll. The size of the data is about 50M, and we use around 1M high-quality license-safe data to fine-tune our model. The filter is created based on aesthetic scores and the motion speed (filtering out both static videos and videos with extreme fast motions). The raw videos are in high resolution (around 2K / 4K), and we downsample them to 480P for the wan2.1 fine-tuning. The downsampling is processed by first cropping to match the aspect ratio, and then resizing.The duration of each video is around 5s to 60s, and the average duration is about 20s. During training our 5s autoregressive model, we randomly sample a continuous chunk of 5s video from the videos; and the videos are sampled uniformly from the filtered dataset. The raw FPS of the video is 25~60; we subsample the frames to best fit the 24 FPS. For the aspect ratio, 90% of the videos are landscape video (with H:W < 1) and the remaining are portrait videos (with H:W > 1). We simply drop all portrait videos to match the characteristics of the WAN2.1 base model. The text caption of the video is obtained from open-sourced multimodal model InternVL-2 (https://github.com/OpenGVLab/InternVL).
>
> We value transparency and reproducibility, and while we cannot release the data, we have submitted all training code for our NVS, video diffusion, and language model experiments. The NVS and language model experiments are done on open-soured datasets, and we are happy to provide checkpoints for these experiments as well.
>
>
>
>
> ## Why was LaCT trained on proprietary data when any video diffusion dataset could have sufficed?
>
> This is due to internal legal and compliance requirements. Our company/institution requires explicit legal approval for external video dataset, and the licenses of many popular video datasets are not considered sufficiently clear for this project. For example, recent large-scale datasets like Panda-70M [1] and Koala-36M [2] source most clips from YouTube, which our organization has a strict policy against using for training in this context.
>
>
> [1] Chen, Tsai-Shien, et al. "Panda-70m: Captioning 70m videos with multiple cross-modality teachers." Proceedings of the IEEE/CVF Conference on Computer Vision and Pattern Recognition. 2024.
>
> [2] Wang, Qiuheng, et al. "Koala-36m: A large-scale video dataset improving consistency between fine-grained conditions and video content." Proceedings of the Computer Vision and Pattern Recognition Conference. 2025

---

> ### Author Response · Authors · 2025-11-22
> **First round rebuttal to reviewer YxH9; Part 2**
>
> ## Limited comparison to alternative TTT methods
>
> Direct comparison to Titans and Atlas is unfortunately not feasible at this time:
> 1. Their implementations are not open-sourced.
> 2. Their papers omit key implementation details (e.g., fast-weight initialization, head dimension, hidden dimension of the fast-weight MLP, parameterization of $\eta_t$ in Equation (18) of Titans paper),  which makes faithful reproduction at scale very difficult.
>
> For comparing with original TTT methods.
> We actually did one small scale comparison in Figure 1(d), where we compare the PSNR versus wall-clock training time, with details showing at Appendix E.4.  Here is the TTT uses the open-sourced Thunderkittens(TK) kernel implementation[1].
>
> We did not use the same TK kernel implementation to conduct LM experiments because: this TK kernel implements bidirectional modeling chunk size of 16 or 64 [1], which cannot be used in language model experiments.
>
> We examined all publicly available repositories from the original TTT authors:
>
> 1. Inference only kernel [2], which does not support backward pass, thus cannot be used for training.
> 2. A pytorch reference implementation with extremely low GPU utilization [3], which are not recommended by the author to use for training.  Running this implementation for our 760M LM setting would require ~95 days on 32xA100, compared to ~20 hours for other baselines.
> 3. The author also released the Jax code[4], which they used to train their model on TPUs.  Running a 760M model for 40B tokens would take ~8.9 days on 32×A100, ~9× slower than our PyTorch pipeline, making extensive evals practically infeasible. Moreover, our entire training and evaluation stack is built in PyTorch; porting all experiments to JAX for a full comparison would require substantial engineering effort in addition to the compute cost.
>
> For these reasons, we limited direct TTT comparisons to the NVS setting, where their TK implementation is usable.  We hope this explains why you only see the comparison with original TTT only in Figure 1(d).
>
> [1] https://github.com/test-time-training/ttt-tk
>
> [2] https://github.com/test-time-training/ttt-lm-kernels
>
> [3] https://github.com/test-time-training/ttt-lm-pytorch
>
> [4] https://github.com/test-time-training/ttt-lm-jax
>
>
>
> ## How does LaCT compare against previous TTT methods, given its much higher propensity for scalability?
>
> As noted earlier, as well as  in the reviewer’s summary, LaCT provides substantial hardware-efficiency gains, which directly improve scalability.
>
> Beyond this, we highlight two additional distinctions with previous TTT methods:
>
> (1) Large-chunk updates naturally support non-causal data structures better. Prior TTT and RNN-based methods are almost entirely designed around strictly causal 1D sequences. In contrast, LaCT’s large-chunk updates allow fast-weight updates at the image, video-frame, or set level, making it far more suitable and potentially more scalable for N-dimensional data such as image sets and videos.
>
> (2) LaCT achieves good efficiency in pure PyTorch, without custom kernels (for sure customized kernel will improve efficiency of LaCT further, which we are also working on, and we provide a earlier version of triton kernels for LaCT in the supplementary material under folder `minimal_implementations`.). Modern RNNs (e.g., Mamba/DeltaNet) typically require specialized CUDA kernels to be efficient on GPUs. LaCT reaches strong hardware utilization with only native PyTorch, which significantly democratizes research in new sequence models and TTT.
> We hope this lowers the barrier for others to develop new RNNs and new TTT variants and potentially discover even more scalable approaches.

---

### Official Review · Reviewer_vmAu · 2025-11-01

**Soundness:** 3
**Presentation:** 4
**Contribution:** 3
**Rating:** 8
**Confidence:** 3

**Summary:**

This paper addresses a critical inefficiency in existing Test-Time Training (TTT) methods when applied to long-sequence data. The authors identify that conventional TTT approaches, which utilize very small online mini-batch sizes for frequent weight updates, suffer from extremely low FLOPs utilization on modern GPUs. To overcome this, the paper proposes a novel paradigm called Large Chunk Test-Time Training (LaCT). The core idea is to move in the opposite direction of prior work by using extremely large chunks of tokens (from 2K to 1M) as the basic unit for updating the model's "fast weights". The authors claim that this approach: (1) Improves hardware utilization by orders of magnitude, leading to significant computational efficiency gains. (2) Enables the effective scaling of large, non-linear fast weights, thereby substantially increasing the model's state capacity. (3) Can be implemented with just a few dozen lines of native PyTorch code, obviating the need for cumbersome and error-prone custom kernel implementations and thus democratizing research in this area. (4) Facilitates the easy integration of more sophisticated test-time optimizers, such as Muon.
The authors embed LaCT within a hybrid architecture that also uses windowed attention to capture local dependencies. The effectiveness of this framework is demonstrated on three diverse and challenging tasks: novel view synthesis, language modeling, and autoregressive video diffusion.

**Strengths:**

1. This paper correctly identifies a critical bottleneck in existing TTT methods when applied to long-sequence data, and proposes the LaCT using massive chunks for updates instead of tiny ones.

2. This paper shows thorough and rigorous Experiments. (1) Validation on novel view synthesis (image sets), language modeling (1D sequences), and video generation (high-dimensional sequences) powerfully demonstrates the method's generality. (2) The scale of the experiments, such as handling context lengths of over 1M tokens and scaling to a 14B parameter video model, is remarkable and convincingly showcases the method's scalability. (3) The appendix contains ablation studies that systematically analyze the impact of key design choices, including state size, optimizer choice, linear vs. non-linear fast weights, and chunk-wise vs. per-token recurrence. This analysis greatly strengthens the credibility of the paper's conclusions.

3. The paper is well-organized and clearly written. The figures are highly effective to provide the summary of LaCT's advantages, the basic framework and the design choices.

**Weaknesses:**

1. LaCT fundamentally treats tokens within a large chunk as an unordered set. While the use of windowed attention mitigates this by handling local structure, the inherent limitations of this design choice are not fully explored. For tasks that are extremely sensitive to fine-grained causal ordering, LaCT might struggle.
2. According to the experimental results in the appendix, LaCT's performance on the VBench benchmark for autoregressive video generation does not show a significant advantage over other AR methods.

**Questions:**

1. Did the authors observe any specific failure cases in the experiments that could be attributed to the lack of strict causality within a chunk?

2. How sensitive is LaCT's performance to the chunk size? In practice, how would you advise balancing an increase in chunk size (for more compute intensity) against an increase in state size (for more model capacity and memory)?

---

> ### Author Response · Authors · 2025-11-22
> **First round rebuttal to reviewer vmAu; Part 1**
>
> We thank the reviewer's appreciation and its insightful questions!
>
>
> ## Any failure cases due to lack of strict per-token causality
>
> The reviewer made an insightful observation, and asked a very very intriguing and open research question:  is there some task that are sensitive to fined-grained causal ordering, or do we observed some failure cases in experiments that could be attributed to this.
>
> In our experiments, we did not observe failure cases attributable to the lack of strict per-token causality within a chunk. A likely reason is that, for causal tasks, LaCT always employs causal sliding-window attention inside the chunk, which preserves fine-grained ordering within the window.
>
> This does not imply such tasks do not exist—only that we have not yet encountered them. We view this as an interesting open research direction: identifying domains where per-token causality is indispensable even with local causal window attentions.
>
> We do, however, have included experiments that illustrate how large-chunk updates behave across different data structures. For example:
>
> In Figure 8(b, left), LaCT (blue) with linear fast weight performs slightly worse than linear per-token recurrence (yellow) under the same state size and FLOPs on 1D LM data.
>
> In contrast for novel-view-synthesis experiment in In Figure 8(b, right),large-chunk linear recurrence performs better than per-token recurrence,  even when the per-token recurrences scan through the images twice (one raster scan and one inverse raster scan) and having more FLOPS, because NVS benefits from bidirectional, non-causal context aggregation across the set of images.
>
>
> ## How sensitive is LaCT's performance to chunk size.
>
> **Regarding hardware utilization**
>
> We measured the hardware utilization of the large-chunk TTT, using the gradient descent with momentum as the TTT optimizer (which is used in our language modeling experiments) across different chunk sizes, while keeping the following fixed: batch size = 4; sequence length = 32,768; SwiGLU intermediate multiplier = 2.
>
> The table below reports the achieved TFLOP/s on H100 SXM5 as we vary chunk size and head dimension:
>
> | head_dim ↓ / chunk_size → | **1024** (TFLOP/s) | **2048** (TFLOP/s) | **4096** (TFLOP/s) | **8192** (TFLOP/s) | **16384** (TFLOP/s) |
> |---------------------------|---------------------|---------------------|---------------------|---------------------|----------------------|
> | **512**                   | 87.1                | 147.8               | 283.2               | 321.8               | 347.7                |
> | **1024**                  | 288.9               | 400.9               | 455.6               | 490.2               | 484.6                |
>
> Here is results on A100 SXM4
>
> | head_dim ↓ / chunk_size → | **1024** | **2048** | **4096** | **8192** | **16384** |
> |---------------------------|----------|----------|----------|----------|-----------|
> | **512**                   | 94.9     | 115.3    | 131.5    | 146.5    | 148.9     |
> | **1024**                  | 140.7    | 161.8    | 176.6    | 187.2    | 167.5     |
>
>
> **Regarding performance**: we are running experiments with different chunk sizes and will update you once done.
>
> ##  Advices on balancing an increase in chunk size (for more compute intensity) against an increase in state size (for more model capacity and memory)?
>
> There is no direct tradeoff between chunk size and state size, and they affect different aspects of the model.
>
> Increasing state size generally improve performance (see Figure 7(a)).
> But it increases FLOPs,  and will make the model more expensive.
>
> Chunk size does not change FLOPs (except when using Muon for state optimization). It primarily affects compute intensity and therefore runtime efficiency.
>
> Larger chunks → higher compute intensity → better GPU utilization, up to a saturation point.
> This will saturate because, given a linear fast weight with dimension $D$, denoting the chunk size as $N$, the compute intensity for the memory query is $\frac{N D}{D + 2N}$, which will increase as you increase the chunk size, but bounded by $\frac{D}{2}$ even if you have infinite chunk size.

---

> > ### Author Response · Authors · 2025-11-23
> > **First round rebuttal to reviewer vmAu; Part 2**
> >
> > ## Sensitivity of chunk size for performance
> >
> > Thanks for raising this insightful question. You and reviewer  21cR all asked this question, which I appreciate a lot.
> >
> >  We conducted an ablation study on the LaCT chunk size in both the language modeling and novel-view-synthesis (NVS) experiments. In all experiments, we fix the window-attention size and vary only the LaCT chunk size.  These two experiments shows that the chunk size has minimal impact on performance.
> >
> > **Ablation: Chunk size in language modeling**
> > We repeat the LM setup from the paper with a smaller compute budget: sequence length 16,384 and 20B training tokens (vs. 32k sequence length trained for 40B in Fig. 7–8). We evaluate per-position loss over 2.5B tokens, as done in the main paper. The per-position loss (length 16,384) is divided into 8 equal segments, and we report the average loss per segment.
> >
> > Results below show negligible differences across chunk sizes:
> >
> >
> > | Val loss at Segment | Chunk Size 2048 | Chunk Size 1024 | Chunk Size 512 |
> > | --- | --- | --- | --- |
> > | 1-2,048 | 2.778 | 2.777 | 2.776 |
> > | 2,049-4,096 | 2.690 | 2.688 | 2.689 |
> > | 4,097-6,144 | 2.682 | 2.680 | 2.682 |
> > | 6,145-8,192 | 2.678 | 2.676 | 2.679 |
> > | 8,193-10,240 | 2.676 | 2.674 | 2.677 |
> > | 10,241-12,288 | 2.675 | 2.672 | 2.676 |
> > | 12,289-14,336 | 2.674 | 2.671 | 2.675 |
> > | 14,337-16,384 | 2.673 | 2.671 | 2.674 |
> >
> >
> >
> > **Ablation: Chunk size in Novel-View-Synthesis experiment**
> > We also test chunk size sensitivity on the object-level NVS experiment using a smaller model (8 blocks instead of 14 as in Figure 7, 8 of the paper). Similarly we. train the model with 8 input and 8 output views.  Each image has resolution of $256\times256$, resulting 1024 tokens per image, 16384 tokens per sequence.  We train each model for 167B tokens, and report the novel view PSNR on evaluation set with 8 input images.  Again, performance is largely unchanged across chunk sizes:
> >
> >
> > | Metric \ Chunk-size | 8 images | 4 images | 2 images | 1 images |
> > |--------|---------|---------|---------|---------|
> > | PSNR   | 28.50   | 28.40   | 28.48   | 28.41   |

---

### Official Review · Reviewer_NZ1F · 2025-11-03

**Soundness:** 4
**Presentation:** 2
**Contribution:** 3
**Rating:** 6
**Confidence:** 3

**Summary:**

This paper revisits Test-Time Training (TTT) for long-context modeling and introduces Large Chunk Test-Time Training (LaCT), which updates fast weights over large token chunks instead of small mini-batches. The method aims to improve GPU utilization, support scaling of non-linear fast-weight networks, and integrate window attention for local dependency modeling together with the Muon optimizer for stable fast-weight updates. The authors evaluate LaCT on three modalities: novel view synthesis, language modeling, and autoregressive video diffusion.

**Strengths:**

1. The proposed approach is general and can be readily adapted to a variety of domains and applications.

2. The evaluation across three domains validates its scalability, with the view synthesis task demonstrating near-transformer performance at much higher efficiency.

**Weaknesses:**

There are some minor weaknesses that do not affect my rating but do affect readability. Some parts of the paper are hard to follow, especially for readers unfamiliar with TTT, as several key details and related work are placed in the appendix. A few figures are incorrectly rendered with shading artifacts, which should be fixed.

**Questions:**

1. The authors report 40B-token training for the 760M model and 60B tokens for the 3B model. Have the authors considered scaling experiments with additional training data (~500B-1T) to better evaluate the method’s true language modeling capability? Given the limited token counts, the model’s performance on ARC-c is nearly at random-guess level, which raises questions about the significance of these results.

2. Additionally, given the training setup and results in Table 9, would the 760M model outperform the 3B model if both were trained on the same number of tokens? The current configuration, where the larger model is exposed to only slightly more data, raises the question of whether the observed results reflect true scaling behavior or simply data inefficiency.

---

> ### Author Response · Authors · 2025-11-22
> **First round rebuttal to reviewer NZ1f; Part 1**
>
> We thank the reviewer's appreciation and its constructive feedbacks! Here we address your comments.
>
> > Reviewer: "There are some minor weaknesses that do not affect my rating but do affect readability."
>
> On readability for readers unfamiliar with TTT.
> We agree that the current preliminary description of TTT in Sec. 2.1 is somewhat abstract. It mainly presents the general equations without offering an intuitive example, which can be difficult for readers new to TTT.
>
> We are considering adding a short, self-contained explanation showing how vanilla linear attention can be derived from a TTT-style key–value reconstruction loss (the explanation provided in our response to reviewer gUWh under the title “Linear Attention under the TTT view” at [here](https://openreview.net/forum?id=Tb9qAxT3xv&noteId=UQsuKdne5E)). We believe this would significantly improve accessibility for readers without prior TTT background, and welcome to comment more about this.
>
> > Reviewer: "Several key details and related work are placed in the appendix.
> Could you specify which details you feel should definitely move into the main text?"
>
> For the related works, I kind of agree of moving the two paragraphs on test-time training, and combining chunk attention with recurrence into the main paper.
>
>
> ## Question-1: Have the authors considered scaling experiments with additional training data (~500B-1T) to better evaluate the method’s true language modeling capability?
>
> We completely agree with the reviewer! With training budgets of 40B (760M) and 60B (3B) tokens, it is expected that performance on challenging benchmarks such as ARC-c remains near random-guess levels. This is a limitation we recognize; it is one reason we placed Table 9 in the appendix and emphasized per-position loss and RULER results in the main paper, which we view as more indicative at this scale.
>
> More broadly, this reflects a well-known challenge in evaluating new architectures:
> how to measure their potential without the very large compute, data, and infrastructure costs of full large-scale LM training (hundreds of billions to trillions of tokens, plus SFT/RL post-training). This is even more difficult for long-context models, where strong long-context benchmarks usually require additional context-extension pretraining. This is more challenging for academia labs than industrial labs.
>
>
> Given these constraints, our goal in the language modeling experiments is not to claim state-of-the-art LM quality, but to:
> 1. Demonstrate that large chunk recurrence (which are not natively designed for 1D causal sequences) with nonlinear RNN states can be trained stably on pure 1D causal LM data.
> 2. Show that, at modest scale, large-chunk nonlinear recurrence can achieve performance comparable to per-token linear recurrence.
>
> This result is already somewhat surprising, as some practitioners assume that per-token recurrence is necessary for 1D causal LMs. Showing that large-chunk recurrence, ignoring per-token ordering within chunk, works in this regime is an interesting finding on its own.
>
> To obtain a cleaner signal about long-context behavior under constrained compute, we also use novel view synthesis (NVS) as a major benchmark, which is shown at the first subsection in our experiment. NVS is particularly well suited for testing long-conext behavior because
> 1. It is inherently a long-context, multi-hop retrieval task where the model must retrieve and aggregate multiple relevant patches to render each novel view, using up to 1M input tokens (see the second paragraph of Sec. 4.1).
> 2. Long-context retrieval is believed to be a major bottleneck for RNN-style models with limited state size [1], so NVS effectively “stress tests” the memory and retrieval capabilities of different methods.
>
> This benchmark offers good signals regarding long-context multi-hop retrieval. I also need to clarify, NVS does not evaluate complex in-context reasoning capabilities. However, as noted above, probing reasoning realistically would require substantially larger pretraining and post-training budgets than we currently have access to. We will clarify this motivation more explicitly in the paper.
>
> [1] Arora, Simran, et al. "Simple linear attention language models balance the recall-throughput tradeoff." International Conference on Machine Learning. PMLR, 2024.
>
>
> ## Question-2: would the 760M model outperform the 3B model if both were trained on the same number of tokens
>
> This is an excellent and subtle question. I will do a 760M training run with the same amount of data as the 3B one, and update when it's done.
> I feel it's coupled, both more data, larger state size, and large model size helps.
> Also, regarding the scaling behavior, I am more interested in scaling over the state size, which we showed brefily in Figure 7(a) and figure 1(c).

---

> > ### Author Response · Authors · 2025-11-28
> > **Update on Question-2: would the 760M model outperform the 3B model if both were trained on the same number of tokens**
> >
> > To address this question, we upgraded the 760M-parameter model to match the training configuration of the 3B model. Specifically, we increased its chunk size and window-attention size to 4096 and trained it on 60B tokens, the same total number used for the 3B model as in Table-9.
> >
> > Here is the results
> >
> > | Model        | S-NIAH-1 4K | 8K  | 16K  | 32K  | S-NIAH-2 4K | 8K  | 16K  | 32K  | S-NIAH-3 4K | 8K  | 16K  | 32K |
> > |--------------|-------------|-----|------|------|-------------|-----|------|------|-------------|-----|------|------|
> > | **Ours-3B (Muon)** | 98.8 | 99.2 | 98.6 | 93.4 | 100 | 99.0 | 83.2 | 30.8 | 95.4 | 90.8 | 55.6 | 19.8 |
> > | **Ours-760M (Muon)** | 100 | 96 | 91.6 | 73 | 100 | 88.4 | 45.4 | 14.6 | 76.6 | 46.6 | 20.8 | 10 |
> >
> >
> > As shown above, the 3B model consistently outperforms the 760M model across nearly all sequence lengths and all S-NIAH settings. This suggests that model capacity plays a meaningful role in long-context needle-retrieval performance, and that the stronger results of the 3B model cannot be attributed simply to differences in token count.

---

### Official Review · Reviewer_gUWh · 2025-11-06

**Soundness:** 3
**Presentation:** 2
**Contribution:** 2
**Rating:** 4
**Confidence:** 3

**Summary:**

In this work the author presents LaCT, Large Chunk Test Time Training. This work tries to solve the problem by TTT frameworks where they operate under extreme low FLOPSs (<5%) utilization on the modern GPUs due to low mini-batches (16 or 64 tokens). This work produces large chunk updates consisting 2k-1M tokens across different tasks. The authors validates their claim on three different tasks a) novel view synthesis (NVS), b) language modeling c) auto-regressive video diffusion.

**Strengths:**

1. This particular method enables large chunk updates enabling higher FLOPs utilization.
2. The authors claims to implement the LaCT layer simply using PyTorch without custom CUDA kernels.
3. The authors present metrics highlighting the throughput.
4. The empirical results shows the validity of the method on various tasks requiring long context updates.

**Weaknesses:**

1. One of the major concerns for me is how is the work different from the Fast Weight Programmer and Expressive RNNs. In both the cases the objective is to store the short term memory as the fast weights, next the hidden state in Expressive RNNs [A] basically are used as the fast weights similar to this work. Next the synaptic weight modifications in Expressive RNNs uses a network that uses gradient descent while the update in this particular work is by calculating a gradient $ g $  from a self-supervised loss $ \mathcal{L} $ over the entire chunk. Thus both this work uses update by a learning algorithm Although this particular work uses MUON as an optimizer.
2.  Can the authors clarify that this method is not compared against the Mamba based LM models, as well as the use of the other architectures that handles long context. The authors also might try to use memory-augmented attentions.
3. The positional embedding interaction with the fast weights seems ambiguous, authors need to clarify this for the fast weight layer.
4. It is not very clear how does the authors address the alignment problem, of TTT. The self-supervised loss is not perfectly aligned with the main task, the gradient can be harmful.

Minor Concerns:
1. The paper needs improvement in the writing, it is quite difficult to follow the flow of the paper.
2. The paper should have compared against other methods such as the current state  RNNs etc.
3. Few of the abbreviations are assumed to be known, the authors should explicitly state them eg: SMs in Section 2.2.

[A]. https://arxiv.org/pdf/2407.04620

**Questions:**

Most of the major weaknesses presented are questions. If the authors can clarify these questions, then I will gladly increase my rating.

Why the related work and many other ablations presented in the appendix?

---

> ### Author Response · Authors · 2025-11-22
> **First round rebuttal to reviewer gUWh;  Part 1**
>
> We appreciate the reviewer's efforts and the constructive feedback regarding the distinction from prior work and clarification on model details. We address your specific questions below.
>
>
> ## How is the work different from the Fast Weight Programmer and Expressive RNNs.
>
>
> The fast weights, recurrent states (in LSTMs, or modern RNNs such as Linear Attention, Mamba/DeltaNet), and TTT-style fast-weight adaptation all fall under the same conceptual umbrella. They are all updated online to store in-context information. For example, “Linear Transformers Are Secretly Fast Weight Programmers” [1] shows that vanilla linear attention and DeltaNet can both be interpreted as fast weight programmers.
>
> Our work also falls within this fast-weight perspective and more specifically follows the test-time training (TTT) framework of Expressive RNNs [2].In this view, one designs RNNs whose state-update (fast-weight update) equations are derived from online gradient descent on a virtual self-supervised loss.  Test-time training opens a vast and principled design space for new generations of RNNs.
>
> I understand this sounds a little bit abstract for people lacking background,I will include an illustrative example in another post for you, showing that vanilla linear-attention can be derived from test-time training framework.  The name of the post would be: "Linear Attention under the TTT view", at [here](https://openreview.net/forum?id=Tb9qAxT3xv&noteId=UQsuKdne5E)
>
> Our main contribution relative to prior fast weights / RNNs / TTT models is the large-chunk update regime: under the TTT framework, we propose updating the fast weights/state once per large chunk of tokens (2K–1M tokens), whereas most prior works update per token or use very small chunks by default. We show that this design has several advantages:
>
> (1) **Higher compute intensity and GPU efficiency**. Large chunks improve compute intensity (Eq. 3 in our paper), which directly translates into higher FLOPs utilization on GPUs.
>
> (2) **More parallelism for nonlinear RNNs.**
> Large-chunk updates allow more parallelism than small-chunk nonlinear updates, leading to better GPU utilization and enabling some degree of context parallelism across GPUs (last paragraph of Sec. 2.2 and Sec. 3.4). This is, to our knowledge, the only practical way to obtain sequence-dimension parallelism for nonlinear RNNs.
>
> (3) **Better support for N-dimensional data.**
> Per-token recurrence enforces strict causality and is tailored to 1D sequences. For other modalities such as image sets (Sec. 4.1) and videos (Sec. 4.3), we often need (locally) bidirectional dependencies between elements. Large-chunk updates naturally fit this: tokens within the same chunk can effectively “bidirectionally attend to each other,” as illustrated in Fig. 3. We in Fig 1(d), comparison with original TTT and Mamba, that such 1D causal sequence model are inferiors for task like novel view synthesis which requires bidirectional modeling.
>
> (4). **No custom kernels; faster research iteration**
> Modern RNNs like Mamba or DeltaNet typically require custom CUDA kernels for efficient execution, which slows down research iteration. In contrast, our large-chunk design allows fast-weight/state updates to be implemented purely in PyTorch, even with more complex update rules (e.g., using Muon as in our experiments). This greatly democratizes research and makes it easier to prototype new RNN-style architectures.
>
> The efficiency advantages are reflected in your summary; here we clarify and expand on points (2)–(4) as the core conceptual difference from prior fast-weight and TTT works.
>
>
> [1] Schlag, I., Irie, K., & Schmidhuber, J. (2021, July). Linear transformers are secretly fast weight programmers. In International conference on machine learning (pp. 9355-9366). PMLR.
>
> [2]  Sun, Yu, et al. "Learning to (Learn at Test Time): RNNs with Expressive Hidden States." Forty-second International Conference on Machine Learning.

---

> > ### Author Response · Authors · 2025-11-22
> > **First round rebuttal to reviewer gUWh; Part 2**
> >
> > ### Comparison against other LM architecures that handles long context, e.g. Mamba.
> >
> >
> > Linear-attention–based architectures are a major family of language models that try to handle long context efficiently.  Common linear attention variants include:
> >
> > - Hebbian-style updates: vanilla Linear Attention, Mamba2, Gated Linear Attention (GLA), etc.
> > - Delta update rule variants, lile DeltaNet, Gated DettaNet, etc.
> >
> > We did comparison with both two type of variants, specifically Gated Linear Attention(GLA) from the Hebbian-style and DeltaNet from the Delta Update rule style.  For a fair comparison, where we also agument each RNN with a sliding window attention of the same size as ours.
> >
> > For the specific question about Mamba, we didn't compare with it in our language model experiments is that the update equation of Mamab2 and GLA are very similar.
> >
> > Both of their state update equation can be written as
> > $$
> > S_{t+1} = G_t \odot  S_t{-1} + v_t k_t^t
> > $$
> > where A gate matrix $G_t$ modulates the previous state element-wise. The main difference is how the gate is parameterized:
> > $$
> > \begin{align*}
> > \text{Mamba2} \quad \quad & G_t = \gamma_t  \mathbb{1}^T \mathbb{1}, \quad  & \gamma_t \in R \\\\
> > \text{GLA:} \quad \quad & G_t = \alpha_t^T  \mathbb{1},  \quad  & \alpha_t \in R^{d}  \\
> > \end{align*}
> > $$
> > Mamba2 uses a scalar gate, while GLA uses a d-dimensional gate. Prior work [3] (Fig. 12) shows that these two update rules yield very similar behavior. Therefore, our comparisons with GLA are representative of the same class of long-context RNN architectures that includes Mamba2.
> >
> >
> > ### "The positional embedding interaction with the fast weights seems ambiguous, authors need to clarify this for the fast weight layer."
> >
> > In all three tasks, we use the same positional encoding scheme as the corresponding softmax-attention baselines:
> >
> > Novel view synthesis: none of the baselines, including ours, use positional encodings.
> >
> > Video generation and language modeling: we apply rotary positional embeddings to queries and keys in the fast-weight layer, exactly as in the full-attention baseline.
> >
> > We will clarify this explicitly in the revised manuscript.
> >
> >
> > ### "It is not very clear how does the authors address the alignment problem, of TTT. The self-supervised loss is not perfectly aligned with the main task, the gradient can be harmful."
> >
> > This in general is not a problem.
> >
> > In the TTT-based sequence modeling literature, the TTT loss primarily defines the associative memory behavior rather than approximating the main task loss. Existing TTT-style models (including Expressive RNNs) commonly use a key–value reconstruction loss: reconstructing $v_t$  from $k_t$. Minimizing this loss encourages the state to implement a key–value associative memory. Softmax attention can also be interpreted as a form of associative memory that stores explicit key–value pairs.
> >
> > In this sense, the TTT loss controls the implementation details of the memory mechanism, and different TTT losses give rise to different sequence models. It is not standard practice to require the TTT loss to be perfectly aligned with the main task loss, and empirically we have not observed optimization problems from this in our experiments.
> >
> > I also provided more details on the paragraph of "Linear Attention under the TTT view:".  I understand that paragraph is a little bit long and apologize for the increased efforts from you.
> >
> >
> >
> > ## Minor questions:
> >
> > > "The paper should have compared against other methods such as the current state RNNs etc."
> >
> > We compare against several modern RNNs. Specifically, we compare with Mamba2 and the original TTT formulation in Fig. 1(a), (b), and (c) of our paper, as well as DeltaNet and Gated Linear Attention (both modern RNNs) in our language modeling experiments.
> >
> >
> > > " Few of the abbreviations are assumed to be known, the authors should explicitly state them eg: SMs in Section 2.2."
> >
> > Thank you for pointing this out. “SM” stands for Streaming Multiprocessor on GPUs. An SM contains tensor cores, CUDA cores, special function units, registers, shared memory, L1 cache, and warp schedulers; it is where the main computation on the GPU is performed. Modern GPUs (A100, H100, B200, etc.) typically have over 100 SMs. We will define this clearly in the main text.
> >
> >
> > > " The paper needs improvement in the writing, it is quite difficult to follow the flow of the paper."
> >
> > I do appreciate the feedback from the reviewer. I can tell that we need include more background and details about modern RNNs, linear-attention and test-time training. If you find the “linear attention under the TTT view” explanation helpful, we are happy to incorporate a concise version of it in the paper.
> >
> > [3] Allen-Zhu, Z. (2025, May). Physics of Language Models: Part 4.1, Architecture Design and the Magic of Canon Layers

---

> > > ### Author Response · Authors · 2025-11-22
> > > **First round rebuttal to reviewer gUWh; Part 3;   Linear Attention under the TTT view.**
> > >
> > > This part is optional to read. We provide it as additional background to illustrate how TTT can be used to derive RNNs by reinterpreting vanilla linear attention (one of the simplest modern RNNs) through the TTT perspective. If you found this section helpful for you to understand the background our paper, we are happy to include them in our paper.
> > >
> > >
> > > Linear attention removes the softmax from standard attention. For a sequence of input tokens,  each token is split to a triplet $(q_t, k_t, v_t)$ (query, key, value) at time $t$. Then the output at time $t$ is:
> > >
> > > $$
> > > o_t = \sum_{i=1}^t (q_t^T k_i) v_i  \in \mathcal{R}^d
> > > $$
> > >
> > > In recurrent form, linear attention can be written as an RNN with a matrix state $S$:
> > >
> > > $$
> > > \begin{align*}
> > > \text{State Update:}  &\quad S_t = S_{t-1} + v_t k_t^T     \quad\quad \in \mathbb{R}^{d \times d} \\\\
> > > \text{Memory Readout:}&\quad o_t = S_t q_t             \quad\quad\quad\quad\quad\; \in \mathbb{R}^{d}
> > > \end{align*}
> > > $$
> > >
> > > Here:
> > >
> > > - $S_t$ represents the recurrent state, it’s a matrix, the cumsum of rank-one matrixs.
> > > - $d$ denotes head dimension.
> > >
> > > -Now comes the key interepretation: think of $S$ not just as a “state” but as **the trainable weights of a linear layer**.
> > >
> > > The update rule can be seen as one step of gradient descent on the negative dot product loss:
> > >
> > > $$
> > > \text{TTT Loss:} \quad \quad  \mathcal{L}_t(S)     = -\langle S k_t , v_t \rangle
> > > $$
> > >
> > > and one step of gradient descent is exactly the same state update equation for linear attention:
> > >
> > > $$
> > > \text{Grad Descent:} \quad  S_t = S_{t-1} - \nabla \mathcal{L}(S) = S_{t-1} + v_t k_t^T
> > > $$
> > >
> > > The **memory readout** is then just a forward pass of this linear layer.
> > >
> > > This viewpoint is actually interesting:  the fixed-size matrix state $S_t$ becomes the fast weight of a linear layer, trained via gradient descent at test-time to memorize the mapping from key to values:  $k_t \rightarrow v_t$
> > >
> > > This loss has some interesting meaning:  it enforces alignment between the output of the fast weight function and the corresponding values—turning the system into a **key–value associative memory**. i.e. associate keys to its corresponding values.
> > >
> > > Importantly, this TTT loss is not the main task loss; it is used to instantiate the associative memory mechanism. Different TTT losses give rise to different sequence models, without requiring direct alignment with the downstream task loss.

---

### Author Response · Authors · 2025-12-03
**Summary**

We thank all reviewers, area chairs, and PCs for their time and effort.

We briefly summarize our contributions below.

We build on the test-time training (TTT) framework to design a new generation of RNN-style models, where the state update can be interpreted as applying gradient descent to a subset of the model’s weights (“fast weights”).

We address a key limitation of existing TTT layers: extremely low hardware utilization caused by low arithmetic intensity and limited parallelism when updating fast weights every few tokens (e.g., 16–64). Our solution is Large-Chunk Test-Time Training (LaCT), which updates the fast weights/state once per large chunk of tokens (2K–1M), instead of per token or tiny chunks.

We evaluate this large-chunk design on three tasks with different data structures:

- novel view synthesis from 2D image sets,

- language modeling on 1D causal sequences, and

- autoregressive video generation on 2D frame sequences.

This design offers several advantages:

- Higher compute intensity and GPU efficiency. Large chunks significantly increase arithmetic intensity(Eq. 3) and FLOPs utilization.

- More parallelism for nonlinear RNNs. Updating fast weights once per chunk introduces parallelism that is otherwise hard to obtain for nonlinear RNNs, improving GPU utilization and enabling some degree of context parallelism across GPUs (Sec. 2.2, Sec. 3.4).

- Better support for N-dimensional data. Per-token recurrence enforces strict 1D causality, which is ill-suited for image sets and videos that benefit from (locally) bidirectional context. Large-chunk updates naturally fit these N-dimensional, partially bidirectional settings.

- No custom kernels; faster research iteration. Unlike many modern RNNs (e.g., Mamba, DeltaNet) that require custom CUDA kernels, LaCT’s fast-weight/state updates are implemented purely in PyTorch, even with more complex update rules such as Muon. This lowers the barrier to experimentation and democratizes research on new RNN-style architectures.

For better reproducibility, we include both training and inference code in the supplementary material.

We thank again for the reviewer's feedback, and we have addressed each reviewer’s comments in detail below.

---

### Meta-Review · Area_Chair_wgG4 · 2025-12-28

**Summary:**

The reviewers generally found the paper to present a clear and practically useful framework for improving the hardware efficiency and scalability of Test-Time Training. The proposed Large-Chunk TTT (LaCT) design was viewed as effective and well-motivated, enabling high GPU utilization and larger state capacity without relying on custom CUDA kernels. The approach was validated across multiple modalities and scales. The submission received initial reviewer scores of 8, 6, 6, 4, and 4. Based on the reviews and the rebuttal, I recommend Accept.

**Reviewer Concerns:**

The rebuttal largely addressed the main technical concerns regarding novelty, design choices, and scalability, clarifying the shift from fine-grained per-token updates to large-chunk updates and their implications for efficiency and modeling flexibility. Additional analyses helped resolve questions about scaling behavior and sensitivity to chunk size. For the final version, many reviewers noted that the paper would benefit from clearer presentation. The authors should improve readability, provide more accessible background explanations for readers less familiar with Test-Time Training, and move important background information and key experimental results from the appendix into the main text.

**Reviewer Scores:**

This submission received initial scores of 8, 6, 6, 4, and 4. Based on the reviews, the rebuttal, and the initial average score, I recommend Accept.

---

### Decision · Program_Chairs · 2026-01-26

Accept (Poster)